# Lava flow hazard map of Piton de la Fournaise volcano

Magdalena Oryaëlle Chevrel[1], Massimiliano Favalli[2], Nicolas Villeneuve[3,4,5], Andrew J. L. Harris[1], Alessandro Fornaciai[2], Nicole Richter[3,5,6], Allan Derrien[3,5], Patrice Boissier[3,5], Andrea Di Muro[3,5], Aline Peltier[3,5]

[1]Université Clermont Auvergne, CNRS, IRD, OPGC, Laboratoire Magmas et Volcans, F-63000 Clermont-Ferrand, France.
[2]Istituto Nazionale di Geofisica e Vulcanologia (INGV), Via Battisti, 53, 56125 Pisa, Italy.
[3]Université de Paris, Institut de physique du globe de Paris, CNRS, F-75005 Paris, France.
[4]Université de La Réunion, Laboratoire Géosciences Réunion, F-97744 Saint-Denis, France.
[5]Observatoire Volcanologique du Piton de la Fournaise, Institut de physique du globe de Paris, F-97418 La Plaine des Cafres, France.
[6]Helmholtz Centre Potsdam, German Research Centre for Geosciences (GFZ), Telegrafenberg, Potsdam, 14473, Germany

*Correspondence to*: Magdalena Oryaëlle Chevrel (oryaelle.chevrel@ird.fr)

**Abstract.** Piton de la Fournaise, situated on La Réunion Island (France), is one of the most active hot spot basaltic shield volcanoes worldwide, experiencing at least two eruptions per year since the establishment of the observatory in 1979. Eruptions are typically fissure-fed and form extensive lava flow fields. About 95 % of some ~250 historical events (since the first confidently dated eruption in 1708) have occurred inside an uninhabited horse-shoe shaped caldera (hereafter referred to as the Enclos), which is open to the ocean on its eastern side. Rarely (12 times since the 18th century), fissures have opened outside of the Enclos where housing units, population centers and infrastructure are at risk. In such a situation, lava flow hazard maps are a useful way of visualizing lava flow inundation probabilities over large areas. Here, we present the up-to-date lava flow hazard map for Piton de la Fournaise based on: i) vent distribution, ii) lava flow recurrence times, iii) statistics of lava flow lengths, and iv) simulations of lava flow paths using the DOWNFLOW stochastic numerical model. The map of the entire volcano highlights the spatial distribution probability of future lava flow invasion for the medium to long term (years to decades). It shows that the most probable location for future lava flow is within the Enclos (where there are areas with up to 12 % probability), a location visited by more than 100,000 visitors every year. Outside of the Enclos, probabilities reach 0.5 % along the active rift zones. Although lava flow hazard occurrence in inhabited areas is deemed to be very low (<0.1 %), it may be underestimated, as our study is only based on post-18th century records and neglects older events. We also provide a series of lava flow hazard maps inside the Enclos, computed on a multi-temporal (i.e., regularly updated) topography. Although hazard distribution remains broadly the same over time, some changes are noticed throughout the analyzed periods due to improved DEM resolution, the high frequency of eruptions that constantly modifies the topography, as well as the lava flow dimensional characteristics and paths. The lava flow hazard map for Piton de la Fournaise presented here is reliable and trustworthy for long term hazard assessment, land use planning and management. Specific hazard maps for short term hazard assessment (e.g., for responding to volcanic crises) or considering the cycles of activity at the volcano and different event scenarios (i.e., events fed by different combinations of temporally evolving superficial and deep sources) are required for

further assessment of affected areas in the future – especially by atypical, but potentially extremely hazardous, large volume eruptions. At such an active site, our method supports the need for regular updates of DEMs and associated lava flow hazard maps if we are to be effective in keeping up-to-date with mitigation of the associated risks.

## 1 Introduction

Lava flow hazard maps represent the probability of inundation by a lava flow per unit area. Because such maps show areas that are the most probable to be affected by lava, it is a fundamental tool in lava flow hazard mitigation and in guiding eruption response management. Lava flow hazard maps for basaltic volcanic centers have been previously produced for several highly active volcanoes including Mount Etna in Italy (Favalli et al., 2005; Favalli et al., 2011; Del Negro et al., 2013), Nyiragongo in the Democratic Republic of Congo (Chirico et al., 2009; Favalli et al., 2009), and Mount Cameroon in the Cameroon (Bonne et al., 2008; Favalli et al., 2012). They have also been produced for most ocean island basaltic shields including Mauna Loa and Kilauea on the island of Hawaii (Kauahikaua et al., 1998; Rowland et al. 2005), Lanzarote in the Canaries, Spain (Felpeto et al., 2001), Pico in the Azores, Portugal (Cappello et al., 2015), Fogo in Capo Verde (Richter et al. 2016), and Karthala in the Grande Comore (Mossoux et al., 2019). Having been active with an eruption every nine months since the beginning of the 20[th] century (Michon et al., 2013), producing such a map for Piton de la Fournaise (La Réunion, France) is a particularly pressing need. At the same time, the wealth of data for eruptive events at Piton de la Fournaise means that we can use this location as a case study to further test, develop and evolve methods for effective lava flow hazard map preparation at frequently active effusive centers.

Following widespread damage due to lava ingress into the town of Piton Sainte Rose in 1977, a volcano observatory, the *Observatoire Volcanologique du Piton de la Fournaise* (OVPF) managed by the *Institut de Physique du Globe de Paris* (IPGP), was established on Piton de la Fournaise in 1979. The objective was to install and maintain, in operational conditions, an adequate monitoring network to ensure surveillance of the volcano, thereby improving hazard mitigation. Today, the permanent monitoring network run by OVPF is one of the densest in the world, comprising more than 100 stations including seismometers, tiltmeters, extensometers, GNSS receivers, gas monitoring stations, cameras and weather stations. This network allows OVPF to anticipate, to detect eruption onset and to provide early warning to authorities so that they can organize implementation of the nationally-mandated response plan to be followed by civil protection (the "ORSEC-DSO plan"), a plan which includes provision for evacuation of volcano visitors and resident populations if needed (Peltier et al., 2018, 2020). However, effective hazard management also requires preparation of hazard maps to guide mitigation measures and actions (Pallister et al., 2019). In this regard, an exhaustive database for eruptive events has been built by OVPF through their response to 77 eruptions between 1979 and 2019, all of which have been geophysically, petrographically and physically mapped and measured. The existence of such a database allows application of a robust empirical approach to hazard mapping and planning. Currently this database includes:

i)     An exhaustive lava flow inventory based on detailed mapping over the period 1931–2019 (Derrien, 2019; Staudacher et al., 2016; OVPF database) coupled with lava flow dating (Albert et al., 2020) that allows estimation of eruption and associated lava flow recurrence times;

      ii)    Three high-spatial (25 to 1 m) resolution Digital Elevation Models (DEMs) acquired in 1997, 2010 and 2016 (Arab-Sedze et al., 2014; Bretar et al., 2013; Derrien, 2019);

iii)   A large database for vent distribution to allow robust calculation of the probability density function of future vent opening (Favalli et al., 2009a).

At Piton de la Fournaise, the impact of lava flow hazard has previously been studied by Villeneuve (2000) and Davoine and Saint-Marc (2016), and an initial lava flow hazard map was published in a national report in 2012 (Di Muro et al., 2012). This map is also mentioned in Nave et al. (2016). However, the hazard map was drafted based on a topography

acquired in 1997, and between the publication of the national report in 2012 and the end of 2019, eighteen eruptions occurred resurfacing approximately 19 km$^2$ of land. Thus, regular reassessments of lava flow hazard at Piton de la Fournaise are needed to allow regular updates of maps in this highly active environment where topography is changing annually due to emplacement of new lava flow units. Mapping also needs to be flexible and reactive to the changing hazard situation as new vents and vent distributions become established (Favalli et al., 2009a). The style (channelized versus tube-fed flow), magnitude (total volume

erupted) and intensity (peak effusion rates) of effusive activity will also evolve to influence areas and lengths attained by lava flow fields (Harris and Rowland, 2001; Keszthelyi and Self, 1998; Rowland et al., 2005; Walker, 1973)

To build a hazard map in such a dynamic environment we employed the DOWNFLOW code of Favalli et al. (2005), this being a stochastic model that allows quick, computationally efficient and flexible estimation of the most probable areas to be covered by lava flows (e.g., Favalli et al., 2009b, 2011; Richter et al., 2016). To apply DOWNFLOW, we need to calibrate it

by obtaining the best-fit parameters to reproduce the area of lava flow coverage for selected flows. Building a hazard map, also involves calculation of the lava flow temporal recurrence intervals, as well as vent distributions (Favalli et al., 2009a). We here describe this methodology, step-by-step, and present the resulting up-to-date lava flow hazard map for Piton de la Fournaise. Additionally, we compare three hazard maps derived from three DEMs acquired in 1997, 2010 and 2016 to assess and discuss the effectiveness of this method, and the evolution of the hazard maps resulting from changes in the topography

as expressed in the up-dated DEMs; as will be a common issue at frequently active basaltic volcanoes. First, though, we present the geological setting and eruptive activity at Piton de la Fournaise so as to define the hazard and risk scenarios for this case type example.

## 1.1. Geological setting

La Réunion island is located in the western Indian Ocean, around 700 km east of Madagascar and 180 km southwest of Mauritius. The island is composed of two large shield volcanoes (Fig. 1a): Piton des Neiges, which is dormant, and Piton de la Fournaise which is one of the most active volcanoes in the world (Peltier et al., 2009; Roult et al., 2012). The Piton de la Fournaise edifice is marked by a large horse-shoe shaped caldera, referred to hereafter by its local name "the Enclos" (French

for "enclosure"). The Enclos is an 8 km wide structure, which is open to the ocean to the east and is surrounded by cliffs of up 100 m high to the west, north and south (Fig. 1b). This structure contains any active lava flow to the Enclos limits, with the west to east slope guiding flows towards the ocean. Although it is commonly accepted that the Enclos is the most recent collapse structure at Piton de la Fournaise, its mechanisms of formation and respective ages are still debated. While it may have been formed by a series of collapses and successive landslides (Bachèlery 1981; Merle et al., 2010; Michon and Saint-Ange, 2008; Merle and Lénat, 2003), it may also be the result of only landslides (Duffield et al., 1982; Gillot et al. 1994; Oehler et al. 2004, 2008). The upper part of the Enclos, whose western plateau is located at >1800-1700 m a.s.l., is recognized here as the caldera *sensu stricto* (Michon and Saint-Ange, 2008) and is named the Enclos Fouqué (EF) (Fig. 1b). It is a relatively flat plateau in the middle of which an asymmetric terminal shield has been built to reach an altitude of 2632 m a.s.l. and with slopes of between 15° (on the northern flank) and 30° (on the eastern flank). The summit of the terminal shield is marked by a 1.1 × 0.8 km caldera ("Cratère Dolomieu") that has been formed by recurrent collapses, with the last collapse occurring in April 2007 (Michon et al., 2009; Staudacher et al., 2009; Derrien et al., 2020). To the west, Cratère Dolomieu is edged with a smaller (0.38 × 0.23 km) crater ("Cratère Bory"). The eastern part of the Enclos, i.e., below 1800 m a.s.l., is divided into two areas (most probably formed by successive landslides; Michon and Saint-Ange, 2008) where the slopes are very steep (>30°), a zone hereafter named "Grandes Pentes" (French for "steep slopes"), and a flatter (<10° slopes) coastal area, hereafter named "Grand-Brûlé" (French for "widely-burned"; Fig. 1b).

## 1.2 Effusive activity and risk

As observed on most basaltic shield volcanoes (Dvorak et al., 1983; MacDonald et al., 1983; Tilling and Dvorak, 1993; Walker, 1988), eruptions at Piton de la Fournaise are produced by dyke/sill propagation following preferential paths leading to the concentration of eruptive fissures within, tangential-to or radially-around the summit caldera and along the main rift zones (Bachèlery, 1981). At Piton de la Fournaise, there are three rift zones: the southeast rift zone (SERZ), the northeast rift zone (NERZ) and the north 120° rift zone (N120; Fig.1b). These rift zones are the preferential zones for opening of eruptive fractures and, consequently, are locations where effusive activity is primarily initiated (cf. Dvorak and Dzurisin, 1993). Two other zones with a high concentration of eruptive fissures have also been identified (Michon et al. 2015), these being the South Volcanic Zone (SVZ) and the Puy Raymond Volcanic Alignment (PRVA; Fig. 1b).

Eruptions are mainly effusive and fed by en-echelon fissure sets, with each fissure being a few meters to a few hundred meters long. Activity at the fissures tends to be Hawaiian to Strombolian in style, to feed lava flows that can extend all the way to the ocean. Lava flows tend to be channel-fed to feed extensive compound lava flow fields, but tubes can form during longer-lived eruptions (Coppola et al., 2017; Harris et al., 2019; Rhéty et al., 2017; Soldati et al., 2018). Lava composition is usually transitional, ranging from aphyric basalt to oceanite and midalkaline basalt (Albarède et al., 1997; Lénat et al., 2012), with eruption temperatures in the range 1150-1190 °C (e.g., Boivin and Bachèlery, 2009; Rhéty et al., 2017; Di Muro et al., 2014). Lavas thus have a relatively low viscosity of the order of $10^2$–$10^4$ Pa s upon eruption (Harris et al., 2016; Kolzenburg et al., 2018; Rhéty et al., 2017; Soldati et al., 2018; Villeneuve et al., 2008).

Most effusive eruptions at Piton de la Fournaise occur within the Enclos. This is part of La Réunion National Park and is uninhabited. However, being a major tourist attraction, the Enclos does receive more than one hundred thousand visitors per year. As a result the hiking trails in, and access roads to, the Enclos can receive heavy pedestrian and vehicular traffic, where 129 000 hikers accessed the summit in 2011 (Derrien et al., 2018). The Enclos also hosts the island belt road (national road RN2) which crosses the Enclos for a length of 9.5 km from north to south, and is located at a distance of 800 m inland from the coast and at an altitude of 130-70 m a.s.l. (Fig. 1b). Being the only east coast line of communication, if cut it severely impedes travel between communities in the south and the north of the island (Harris and Villeneuve, 2018). This road had an average traffic usage of more than 4500 vehicles per day in 2014 (INSEE, 2014).

Based on an analysis of records available since the first observations of activity in 1640, Villeneuve and Bachèlery (2006) estimated that 95 % of the historic eruptions have occurred in the Enclos Fouqué caldera. These are, hereafter, termed "proximal" eruptions as they are proximal to the terminal shield. Fissures and associated lava flows produced by these proximal eruptions may cut hiking trails and generate a risk for the visitors (Derrien et al., 2018), and necessitates evacuation and prohibition of access to the Enclos (Peltier et al., 2020). To ensure effective closure of the Enclos, a 2 m-high gate located at the head of the trail that descends a steep, narrow scallop in the Enclos cliff and allows hikers to access the Enclos via the Pas de Bellecombe Jacob (Fig. 1), is physically closed. Fissures may also open on lower part of the Enclos, in the Grandes Pentes or in the Grand-Brûlé, i.e., below 1800 m a.s.l.. Eruptions outside of the Enclos, called *Hors Enclos* eruptions – "hors" being French for outside, are high risk for inhabitants as fissures can open in or above inhabited areas near the coast, such as above the villages of Sainte-Rose and Saint-Philippe (Fig. 1b). They can also open in the inhabited highlands of the volcano, where there are the towns of La Plaine des Cafres and Plaines des Palmistes (Fig. 1b). For example, the eruption of April 1977, which was the first *Hors Enclos* eruption since 1800, caused evacuation of Piton Sainte-Rose, entirely burnt or damaged parts of 26 buildings (including houses, a church, the police station and a gas station), and buried part of the Sainte-Rose municipality (Kieffer et al., 1977; Vaxelaire, 2012). A second *Hors Enclos* eruption in March 1986 caused around 400,000 euros of damage to houses, household contents, agriculture, roads and utilities in the municipality of Saint-Phillippe (Bertile, 1987; Morin, 2012). In 1998, a third eruption occurred outside of the Enclos with lava moving down steep slopes above the village of Bois Blanc, but flow fronts stopped before reaching the inhabited areas (Villeneuve and Bachèlery, 2006).

**1.3 Cycles of effusive activity**

At Piton de la Fournaise, spatiotemporal cycles of eruptive activity can be defined with typical periods of one year to slightly more than a decade. According to Peltier et al. (2009), Got et al. (2013) and Derrien (2019), cycles are driven by superficial processes being controlled by the evolution of the shallow (<2.5 km below the summit) stress field. A cycle typically starts with one or a few summit eruptions, followed by one or a few proximal eruptions (with vents opening on the terminal shield or at its base). A cycle may then end with a large-volume, distal ("low flank") eruption in the Grande Pentes or Grand Brulé area, as was the case in 2007, or with an *Hors Enclos* eruption, as was the case for the cycle that ended in 1977. In addition to these superficial cycles, lava composition seems to be affected by cyclic changes in the "fertility" of the source

magma as observed since 1930, the date since which petrology of the volcanic products have been well documented (Vlastélic et al., 2018). These deep source-related cycles are typically of longer duration (20–30 years) than superficial cycles. They are characterized by a continuous increase of source fertility until reaching a maxima. This peak is followed by a continuous decrease in source fertility, until a minima is reached which marks the end of a cycle and the beginning of a new one (Vlastélic et al., 2018). High-volume distal eruptions associated with pit crater collapse (cf. Walker, 1988), as in 1931-35, 1961, 1986 and 2007, have happened at the three-quarter-stage of a compositionally defined ("deep-seated") cycle, i.e., just after the peak of source fertility (Derrien, 2019).

The average volume of lava emitted per eruption between 1970 and 2007 was about $10 \times 10^6$ m$^3$ (Peltier et al., 2009). In 2007, when the activity was punctuated by a low-elevation (590 m a. s. l.) flank event in the Enclos, 140 to $240 \times 10^6$ m$^3$ of lava was erupted in 30 days (Derrien, 2019; Roult et al., 2012; Staudacher et al., 2009). Historically the 2007 event has been, volumetrically, the largest single effusive event witnessed at Piton de la Fournaise. Through the end of 2019, the March-April 2007 eruption has been followed by a further 25 "typical" eruptions, each with a volume of $5 \times 10^6$ m$^3$ and being fed from vents at higher (> 1700 m a.s.l.) altitudes (OVPF reports ISSN 2610-5101). The activity cycles show that eruptive fissures are more likely to open at lower altitudes and outside of the Enclos towards the end of superficial cycles, and particularly voluminous effusive events may be expected just after the peak in a deep seated cycle. However, such atypical low flank and high-volume events are difficult to predict and can therefore be of high risk.

## 2 Methods and data

### 2.1 Digital Elevation Models

Given the sensitivity of lava flow paths to topography (Favalli et al., 2005), for a target where the topography changes so often as is the case at Piton de la Fournaise, using a frequently updated Digital Elevation Model (DEM) is essential for short term lava flow hazard assessment. In this study we used the most recent DEMs to build the volcano hazard map. However, because we have three DEMs produced at different times we can also build and compare three hazard maps of the Enclos area for three different time periods between 1997 and 2016. The first DEM was obtained for the whole island as derived from Structure From Motion analysis of aerial photographs (i.e., application of stereophotogrammetry) acquired during 1997 campaign of the *National Geographic Institute* (IGN, which since 2012 has been named the *National Institute of Geographic and Forest Information*). This DEM has a 25 m horizontal resolution, and was available on a local projection (Gauss-Laborde Réunion) where elevations are based on the ellipsoid (Villeneuve, 2000). The second DEM was acquired during 2008-2009 by the IGN via airborne Light Detection And Ranging (LIDAR) and has a 5 m horizontal resolution over the whole island, and 1 m near the coast. The vertical resolution is 0.05 to 0.1 m (Arab-Sedze et al., 2014). This DEM was released in 2010 and was resampled at a 5 m resolution across the entire island. It is hereafter referred to as the "2010 LIDAR DEM from IGN". Finally, a most recent DEM was obtained in April 2016 from optical Pléiades satellite images acquired in Stereo Triplet mode

and processed by the *French Centre National d'Etudes Spatiales* using their S2P restitution code (de Franchis et al., 2014). This DEM is restricted to the Enclos, which is where all topographic changes took place between 2009 and 2016. Although the horizontal resolution was 50 cm, it was resampled to 5 m for our study.

## 2.2 Lava flow recurrence time

As a first step in building our lava flow hazard map we need to estimate lava flow recurrence times on the volcano flanks. To do this we prepared a complete, exhaustive and statistically robust eruption inventory (cf. Peltier et al., 2021). For Piton de la Fournaise, the inventory of all historical eruptions starts at the beginning of the 18th century with the first confidently dated eruption of 1708 and can be built using the detailed chronologies in Michon et al. (2013), Morandi et al. (2016) and Villeneuve and Bachèlery (2006). By compiling these studies and convolving them with the record maintained since establishment of the OVPF (Peltier et al., 2009; Roult et al., 2012; Vlastélic et al., 2018), as well as recent OVPF reports (ISSN 2610-5101), we estimate that Piton de la Fournaise had about 250 eruptive events over a period of three centuries. This inventory is given in Table S1.

To produce the lava flow hazard map, we need to account for recurrence of individual lava flow fields and not the number of eruptive events. This is because an eruptive event may produce several lava flow fields that can be located several kilometers apart. Therefore, an individual lava flow field is counted as a discrete entity and entered into the database when it is either temporally or spatially separated from another flow field. Note that in the case of a fissure opening perpendicular to the slope, the lava may erupt uniformly along the fissure to feed several lava flow units simultaneously to form a flow field of many lava fingers (Harris and Neri, 2002; Kilburn and Lopes, 1991, 1988). In such a setting we counted only the main, longest flow, and do not consider all fingers that comprise the compound lava flow field in the database (cf. Walker, 1973). In the case where there is a pause in the eruption and new flows are emitted in a second phase of activity, then the main flow produced after the hiatus is also counted. Finally, if an eruption simultaneously feeds lava flows at multiple, spatially distinct locations, each lava flow site is counted as a separate unit. Following this counting strategy, one eruption may therefore be associated with several lava flows. Since the creation of OVPF (i.e., late-1979) until the end of 2019, a period when the volcano has been continuously monitored so that the inventory can be trusted to be 100 % complete, the volcano erupted 77 times within which we can identify 128 distinct lava flows (Table S1). This translates to around 1.7 lava flows per eruption.

Given the fact that lava flows are more frequent in the Enclos than beyond its limit, we need to count the number of lava flows inside the Enclos based on a different period than for flows outside of the Enclos. For this, we divided the volcano into five regions. The first region was the Enclos, and regions 2 to 4 covered the *Hors Enclos*, which was sub-divided into three regions: the northeast flank (NE), southeast flank (SE) and the highlands along the N120 rift zone (N120). The fifth region included all remaining of Piton de la Fournaise (Fig. 2a). Inside the Enclos, we consider a time period since 1931, this being the date since which a continuous record and reliable mapping has been possible (Derrien, 2019). Until the end of 2019, this involved a total of 193 individual lava flows (Fig. 2a; Table 1; Table S1). The recurrence time of lava flows within the

Enclos is therefore estimated to be one every 5 to 6 months for the 1931-2019 period. Over this period, only three eruptions occurred outside of the Enclos, and six lava flows were counted for these three events (three in 1977, two in 1986 and one in 1998). To calculate the recurrence time of *Hors Enclos* lava flows, six is a rather small number of cases if we are to ensure good statistical representation. We therefore increased the time period to extend back to 1708. Over the 1708–2019 period, fifteen lava flows were registered outside of the Enclos (Fig. 2b; Table 1; Table S1). Nine lava flows occurred on the southeast flank of the volcano. Of these, five were witnessed by inhabitants (in 1774, 1776, 1800 and two in 1986), one was dated at 80 ±35 BP using $C^{14}$ dating on the carbon in the soil below the Piton Raymond lava flow (Vergniolle and Bachèlery, 1982), and three other flows were dated at 1726, 1765 and 1823 by measuring the base diameter of pioneer trees (Albert et al., 2020). On the northeast flank, we counted five flows that were all witnessed (one in 1708, three in 1977 and one in 1998). Within the less active N120 rift zone, the number of historical eruptions is not large. According to Morandi et al., (2016), eleven tephra or lava flow deposits can be dated since 2920±30 BP, but only one lava flow has been dated after 1708 (Fig. 2b; Table 1; Table S1). This eruption (named Piton Rampe 14) was dated by $C^{14}$ at 140±90 BP (Vergniolle and Bachèlery, 1982; Morandi et al., 2016). Another recent eruption is also suggested from some poorly characterized (lapilli) deposits close to the Trous Blancs area at 145±30 BP (Morandi et al., 2016), but no lava flow has been identified associated with this event. However, if we consider the single lava flow at Piton Rampe 14 since 140±90 BP, or the eleven eruptions since 2920±30 BP, the eruption recurrence time does not vary significantly, being one every 263 years or one every 209 years for the two periods, respectively. We therefore assume just one eruption over our 311-year period (1708-2019) of records for the N120 rift zone.

Overall, this inventory represents a minimum bound on eruptive activity because it is possible that other eruptions and lava flows were not observed nor have yet been identified in the geological record and dated. The minimum recurrence of *Hors Enclos* lava flows since 1708 is therefore estimated at one every 21 years (Table 1). The relative occurrence probability is thus calculated by normalizing the number of lava flows per year within each region over the given period. The resulting relative probability of occurrence during the next eruption for a lava flow is 97.8 % inside the Enclos and 2.2 % outside of the Enclos. Beyond the Enclos relative probability can be divided into 1.4, 0.7 and 0.2 % for the southeast, northeast and N120 rift zones, respectively (Table 1).

## 2.3 Probability of vent opening

DOWNFLOW has been previously applied to produce lava flow hazard maps at Mount Etna (Favalli et al., 2009c, 2011), Nyiragongo (Chirico et al., 2009; Favalli et al., 2009b), Mount Cameroon (Favalli et al., 2012), and Fogo (Richter et al., 2016). We here follow the same methodology developed since (Favalli et al., 2009c) and assume that future vents are more likely to open in areas where previous vents cluster. The probability of future vent opening is therefore determined on the basis of location of historical vents but must be scaled to the lava flow recurrence probability within each region, under the general assumption that the characteristics of future eruptions will be similar to those of past eruptions.

Our inventory of vents for Piton de la Fournaise (Table 2, Fig. 3a) is based on the mapped scoria cone distribution (Davoine and Saint-Marc, 2016; Michon et al., 2015; Di Muro et al., 2012; Villeneuve and Bachèlery, 2006) and is here updated with all new vents formed between 2015 and the end of 2019 (Fig. 3a). We counted any scoria cone that is morphologically definable and included undated scoria cones. This method implies that the number of vents over the entire volcano is much higher (726; Table 2) than that of the number of lava flows considered (208, Table 1). Inside the Enclos, for the period post-1931 (Table 3), we did not consider undated scoria cones but, in some places, we counted more than one scoria cone along a fissure although only one lava flow formed. This therefore also results in higher number of vents than lava flows (Table 3). Conversely in the summit craters, we were not able to properly locate all vents, due to burial by subsequent activity, but could map and count the number of flows. This, thus, resulted in an underestimation of number of vents, and a higher number of lava flows than vents in this area.

The vent density distribution (number of vents per unit area) was then obtained by applying a symmetric Gaussian smoothing kernel to the map of vent locations (Bowman and Azzalini, 2003; Favalli et al., 2012; Richter et al., 2016), with a bandwidth that is a function of the local vent density. The vent density distribution is presented in Figure 3a. Within the Enclos, the highest vent concentration is located to the southeast of the summit crater where the density is up to 51 vents per $km^2$. To the east, the concentration decreases from 8 vents per $km^2$ in the Enclos Fouqué caldera, to 3 and 0.5 vents per $km^2$ in the Grandes Pentes and the Grand-Brûlé, respectively. Outside of the Enclos, vent densities range from 0.01 to 8 vents per $km^2$ in the rift zones with a highest concentration of up to 21 vents per $km^2$ within the PRVA (as already noted by Michon et al., 2015).

We find that, of the 726 mapped vents, 45 % (327) are within the Enclos, while 55 % (399) are outside of the Enclos. For the Hors Enclos vents, 20 % (142) are on the N120 rift zone, 13 % (98) are on the southeast flank, 7 % (49) are on the northeast flank, and 15 % (110) are elsewhere on the volcano flanks (Table 2). These distributions of vents per region differ significantly from the lava flow relative occurrence probability per region (Table 1). For example, while the N120 rift zone accounts for 20 % of the total number of vents (Table 2), only 0.2 % of the lava flows recorded since 1708 have occurred in this region (Table 1). Instead, within the Enclos, where ~98 % of the lava flows have been emplaced (Table 1), we count only 45 % of the total number of vents (Table 2). This difference is mainly due to the period of time required for cones located in the various regions to disappear. In the Enclos, resurfacing processes and landscape changes, such as fissure opening, building of new scoria cones over old ones and burial by lava flows occurs at much higher rates than beyond the Enclos. Therefore, the lifespan of scoria cones (vents) within the Enclos is considerably shorter than compared to that of scoria cones outside of the caldera, where ~66 % of the Enclos has been resurfaced at least once since 1931. Moreover, the selected time period (1708-2019) does not cover the full variability in eruption location and activity cycles documented by geological studies (Morandi et al., 2016 and references therein), where (in the geologic past) eruptions outside of the Enclos have, at times, been much more frequent than during the last 300 years. For this reason, the sum of probability density function for future vent opening within a given region of the volcano was set equal to the relative occurrence probability of lava flow in that region, while the spatial distribution for future vent opening within each region follows the vent distribution itself.

The resulting map of the probability density function of vent opening per unit area is given in Figure 3b. The highest probabilities exceed 10 %/km$^2$ with a maximum of 24 %/km$^2$ in the Dolomieu crater and across the proximal area of the terminal shield, while moderate values are obtained across the Grandes Pentes and in the Grand-Brûlé (0.01–0.5 %/km$^2$ and 0.003–0.01 %/km$^2$, respectively). The northeast and southeast rift zones also have low to moderate values (0.003–0.5 %/km$^2$), while the N120 rift zone has a value of less than 0.003 %/km$^2$. For the rest of the volcano the probability of vent opening is
very low, being less than 0.0001 %/km$^2$. However, we note an area of relatively high values (up to 0.5 %/km$^2$) located at the PRVA (Fig. 3b).

## 2.4 DOWNFLOW model calibration

Lava flows are gravitational flows that follow approximately the steepest descent path defined by the underlying topography (Harris, 2013). DOWNFLOW is a numerical code that computes a number ($N$) of steepest descent paths from a
given point over a DEM that is modified by randomly applying a vertical perturbation in the range of $\pm \Delta h$ at every pixel (Favalli et al., 2005). By iteration over $N$ runs, the code computes whether a pixel is "hit" by a lava path or not. The result of a simulation represents the probabilistic estimation of the lava flow inundation area for the given $\Delta h$, $N$ and DEM combination, regardless of the lava properties. Thus, before applying DOWNFLOW to a given case, the key input parameters $\Delta h$ and $N$ need to be defined for the volcano and DEM in question (Favalli et al., 2005, 2012; Richter et al., 2016). Calibrating DOWNFLOW
therefore consists of finding the parameters $\Delta h$ and $N$, for a given DEM, that are able to best fit the model-generated and actual lava flow areas (Favalli et al., 2005). To do this at Piton de la Fournaise, a range of $\Delta h$ (0 to 5 m) and $N$ (100 to 10000) were applied to selected lava flows. For each DEM, the lava flows were selected if they occurred on the unmodified DEM (i.e., the underlying topography is known). DOWNFLOW then computes the array of steepest descent paths out to the limit of the DEM which, in our case, is the coast, but does not allow computation of lava flow lengths. Therefore, for the calibration exercise,
the simulations were cut at the actual length of the flow under consideration (Fig. 4). Following Favalli et al. (2009b) and Richter et al. (2016), the best fit parameters can be found by comparing the actual, mapped lava flow area ($A_R$) with that generated by the simulation ($A_S$):

$$\mu = \frac{A_S \cap A_R}{A_S \cup A_R},$$   (1)

Under this condition, $\mu$ is a measure of the "goodness of fit" between simulated and actual parameters, where if $\mu = 1$ then the
two areas coincide perfectly and if $\mu \rightarrow 0$ then the simulation becomes increasingly unrealistic. Best fit parameters are usually obtained for $\mu \cong 0.5$ (Tarquini and Favalli, 2011). Proietti et al. (2009) and Spataro et al. (2004) evolve this approach slightly by considering a fitting function of $e_1 = \sqrt{\mu}$. This yields the same results, but gives numerical values closer to one.

We performed three calibrations, one for each of the three available DEMs (Figure 4). In total, we ran 70,000 simulations. For the calibration based on the 1997 DEM, that has a resolution of 25 m, we selected the five flows that were
erupted between 1998 and 2007 and obtained a best fit of $\mu = 0.50$ for N > 6000 and $\Delta h$ > 4 m. For the calibration based on

the 2010 and 2016 DEMs, that were both set at 5 m pixel resolution, we considered four and six flows between 2010–2015 and 2016–2019, respectively (all emplaced on the unmodified topography). The best fit was obtained at μ = 0.54 and μ = 0.51, respectively, for N > 5000 and $\Delta h$ of around 2 m (Fig. 4). Note that the difference in DEM resolution (25 m for 1997 and 5 m for 2010 and 2016) implies that the random noise in elevation ($\Delta h$) is applied on a different spatial frequency. On a given topography, the higher the pixel size the greater is the amount of random noise that needs to be applied. This means that for lower resolutions we expect that the best fit will be obtained for higher values of $\Delta h$. The calibration parameters chosen to run DOWNFLOW and build the hazard maps was thus N = 10,000 for all DEMs, but $\Delta h$ = 5 m for the 1997 DEM and $\Delta h$ = 2 m for the 2010 and 2016 DEMs.

**2.5 Lava flow length**

To properly evaluate the probability of lava flow inundation, an estimate of expected lava flow lengths is required. Several methods exist to estimate the most likely length of a lava flow. At Etna, Favalli et al. (2005) observed a linear relationship between the altitude of the main vent and the maximum possible length of the associated flow. Another method is based on the empirical relationship between the average effusion rate during an eruption (i.e., mean output rate, MOR: total volume erupted divided by the duration of the eruption; Harris et al., 2007) and the length of the flow (Walker, 1973). Alternatively, expected cooling-limited flow lengths can be calculated using a thermo-rheological model, such as FLOWGO (Harris and Rowland, 2001). This approach uses theoretical flow cooling and crystallization properties to estimate the point at which forward motion is no longer rheologically possible (Rowland et al., 2004; Wright et al., 2005). Here, at Piton de la Fournaise no clear relationship was found between lava flow length and vent elevation. Due to the great changes in slope within the Enclos (<3° in the Enclos Fouqué to 35° in the Grandes Pentes area and then back to <10° at the coast), and the presence of deep valleys on the volcano flanks, no clear relation was found between MOR and lava flow length. Neither was it possible to simulate a relationship by obtaining run out lengths over a range of effusion rates using FLOWGO, as each slope condition, or changes between different conditions, changes the relationship. However, given the high number of lava flows, it was instead possible to compute the probability that a pixel at a given distance from the vent will be "hit" by lava from the lava flow length frequency distribution of the historical flows. To generate this probability function, we measured the long axis (as a proxy for the length) of all counted lava flows (Fig. S1). Inside the Enclos, the lava flow length distribution frequency is obtained for all the mapped flows since 1931 and until 1997, 2010 and 2016, as well as until the end of 2019 (Fig. 5a). For the *Hors Enclos* lava flows, we extended the database to all flow units that have been mapped by Bachèlery and Chevallier (1982), even if they are not dated. This gave us a total of 43 lava flows to work with (see also Di Muro et al., 2012; Principe et al., 2016; Fig. 5a, Fig. S1). For each time period, the number of lava flows reaching a given length was then converted into a lava flow length probability distribution in terms of %/km. From this probability distribution, the probability that a point at any given downslope distance from a vent will be reached by lava can be calculated (Fig. 5b).

## 2.6 Building probabilistic lava flow hazard maps

A lava flow hazard map gives the probability, at each point, of inundation by the lava upon the occurrence of the next effusive eruption from any given point (Rowland et al., 2005; Wright et al., 2008). To produce our hazard map, DOWNFLOW was thus run at each vent point in a grid of computational vents with a 100 m cell size. For the whole of the Piton de la Fournaise edifice, this represents a total of 126,000 vents and simulations, of which 12,000 were inside the Enclos. The resulting database of simulations provides an inundation matrix (or mask) in which each cell is assigned probability $P_{ij}$, where $P_{ij} = 1$ if pixel $i$ is hit by the simulation of a lava flow originating from vent location $j$, and $P_{ij} = 0$ otherwise (irrespective of the distance between $j$ and $i$). The hazard at any pixel $i$, with a given size $(\Delta x; \Delta y)$, is defined as the total probability $H_i$ that pixel $i$ may be inundated by lava originating at any possible vent location $j$ (Favalli et al., 2012):

$$H_i = \sum_j \rho_{Vj} \Delta x \Delta y \cdot P_{ij} \cdot P_{Lij} , \tag{2}$$

This sum extends over all possible vent locations $j$ with the coordinates $x_j$ and $y_j$, and $\rho_{Vj}$ is the probability density function of a vent opening at location $j$, as shown for the whole edifice in Fig. 3b. In addition, $P_{Lij}$ is the probability that a lava flow originating from vent $j$ will reach pixel $i$ along the calculated flow path (black curve in Fig. 5b). Using this methodology, we produced a hazard map for the entire volcano, plus three hazard maps for the Enclos area: one for each of the three DEMs (that is for 1931-1997, 1931-2010 and 1931-2016).

To be as up-to-date as possible, the hazard map for the entire volcano was derived using a combination of the 2016 DEM for the Enclos area and the 2010 DEM for the rest of the volcano (Fig. 6). We can assemble these two DEMs because the 2010 DEM will not have been affected by topographic change outside of the Enclos, as there were no *hors Enclos* eruptions between 2010 and 2016. We run DOWNFLOW using the best fit parameters as determined for the 2016-2019 period (Fig. 4c). In the Enclos, the lava flow length distribution was determined over 1931-2019, while for rest of the volcano, we considered the length of the flow units mapped by Bachèlery and Chevallier (1982) (Fig. 5, Fig. S1). This is then convolved with the probability density function of vent opening from all the historical vents presented in Figure 3b following Equation 2. To build the successive hazard maps of the Enclos, we take the appropriate data set from the time of DEM generation back to 1931 (Table 3). The data set includes: (i) lava flow length probability distribution ($P_{Lij}$) extracted from the lava flow length measurements (Fig. 5), (ii) the probability of future vent opening ($\rho_{Vj}$) as based on the vent distribution and the corresponding recurrence time of lava flows, and (iii) the DOWNFLOW mask ($P_{ij}$) derived from the calibration values ($N$ and $\Delta h$) tailored for each case (Fig. 4).

## 3. Hazard maps

### 3. 1 Hazard map for the entire volcano

Our lava flow hazard map for Piton de la Fournaise is presented in Figure 6. The map clearly shows that the highest probability of lava flow inundation for the next eruption at Piton de la Fournaise is located within the Enclos. This high probability of lava flow invasion in the Enclos is, of course, due to the high frequency of eruptions in this zone in comparison to the rest of the volcano. Indeed, the probability of lava inundation is calculated to be high (>1 %) for about half (47 %) of the Enclos area. The probability of lava flow invasion reaches extremely high value (12 %) at the summit craters, and in some places within the Enclos Fouqué area (i.e., above 1800 m a. s. l.) and across parts of the Grandes Pentes. In the Grand-Brûlé area, the probability is calculated as being intermediate (up to 2 %) to low (0.1–0.5 %) We note that the there is a relatively high probability (of up to 2 %) that flows will cut the belt road in some places.

The highest probability of lava flow inundation outside of the Enclos is up to 0.5 % and is on the southeast and northeast flanks of the volcano as well as in the PRVA (Fig. 6). The less active N120 rift zone has a very low, but still non-negligible, probability of lava flow invasion (<0.01 %). Although the probability of vent opening is higher at greater altitudes (Fig. 3), the lava flow length is usually longer outside of the Enclos than inside (Fig. 5). This implies that outside of the Enclos the probability of lava flow inundation remains the same as distance increases from the vent, while inside the Enclos it reduces with distance from the vent (n.b. lava flow length distribution peaks are at around 3000 m, Fig. 5). This means that outside of the Enclos the probability distribution for lava flow inundation seems to depend mostly on vent location and the flow path, as influenced by local topography, rather than distance from the vent.

### 3.2 Evolution of the hazard map within the Enclos

The data set used to build the three hazard maps is presented Table 3 and shows that the number of scoria cones increased from 88 for 1931–1997 to 186 by 2019, and the number of lava flows emitted almost doubled from 111 for the period 1931–1997, to 195 for the period 1931–2019. The percentage of vents in the summit craters versus those forming in the rest of the Enclos increased from 3.4% for the period 1931–1997, to 5.8% for the period 1931–2010 and then was roughly the same between 2010 and 2019 (Table 3). The probability of lava flow in the summit craters (Table 3) therefore decreased from 27.9 % in 1997 to 24.1 % in 2019, reflecting that eruptions within the craters became rarer over the analyzed period, being non-existent between 2014 and 2019.

The spatial distribution of lava flow inundation probability for the three considered periods (1931–1977, 1931–2010 and 1931–2019) are presented in Figure 7. The lava flow probability inside the summit craters is not represented in Figure 7 because lava emitted from a vent opening in the summit crater will become entrapped in the pit, and will not to be free to flow down the slopes of the Enclos. Therefore, we did not perform simulations from the summit pit crater area. The three maps present some common features a well as changes over time. The increasing DEM resolution from 2010, where there was a decrease in pixel size from 25 m in 1997 to 5 m in 2010, generally improves the sharpness of the maps. This causes the high

probability areas (red to purple and blue in Figure 7) to be smaller in size in the 2010 and 2016 maps than in the 1997 map. On the three hazard maps, the highest probabilities (2 to >10 %) of lava flow inundation are concentrated above 1800-1700 m a.s.l. (i.e., in the Enclos Fouqué caldera), with the highest values located to the south-east of the terminal cone, along the

420 continuation of SERZ inside the Enclos, and bordering the south wall of the Enclos (Figure 7). Between 1997 and 2010, this high probability area to the south-east of the terminal cone slightly decreases in terms of probability values and size. In contrast, in the 2016 hazard map this area has higher values of up to 12 % and is spread over a larger area than in 2010.

In Figure 8, we give the difference in hazard probability between the hazard maps. The improved DEM resolution between 1997 and 2010 has a direct effect on the spatial distribution of probability, resulting in an overall lowering of the

425 probabilities and reduction in the area of coverage of high probabilities (Figure 8). We note that, on the 2010-1997 map, the noise is partly due to a difference in acquisition methods (LIDAR versus photogrammetry), and on the 2016-2010 map the positive difference near the cost is due to vegetation that has not been removed from the 2016 DEM. One of the main differences in the spatial distribution of lava flow probability between the 2010 and 2016 hazard maps is located to the south of the terminal shield, where the August 2015 lava flow field was emplaced (see Figure 7). This lava flow field has a volume

of $35.5 \times 10^6$ m$^3$ and an average thickness of 8.5 m (OVPF database, Figure 8) and affected the probability calculation by lowering the value in this area. However, although in 2016 hazard map the probability of lava flow invasion was estimated to be low, the 2018 lava flows were emplaced exactly there. This highlights the important effect of vent opening probabilities that remains high in this region and may overcome the effect of topography changes in determining flow paths.

**4. Discussion**

According to Calder et al. (2015), uncertainties in hazard maps are mainly related to three issues:

> "*(i) the incompleteness and bias of the geological record and the extent to which it represents possible future outcomes; (ii) the fact that analyses based on empirical models rely on a priori knowledge of the*

440 > *events; and (iii) the ability of complex computational models to adequately represent the full complexity of the natural phenomena*".

Our hazard maps presented here in Figures 6 and 7 have been produced from available historical and geological records of when and where past lava inundation has occurred at Piton de la Fournaise, as well as vent location and lava flow length. The quality and detail of these records improve with time, and are better inside the Enclos than outside. Secondly, the maps are

445 based on stochastic simulations of lava flow paths using DOWNFLOW, the reliability of which depends on the quality, and up-to-dateness of our topographic model (DEM). This is an issue at frequently active effusive centers, where emplacement of new lava units causes the topography to be in a state of near-constant change. Here we thus discuss the validation of the hazard maps, the uncertainties related to their interpretation, and the extent to which they are adequate in assessing risk associated with future eruptions.

## 4.1 Validating hazard mapping with recent eruptions

In Figure 7, we compare the three maps of the Enclos created for each time period with emplacement location of subsequent lava flows. We see that most lava flows occurred in high probability zone (> 2%, red zone). However, a majority of the flows did not necessarily extend into the very high probability zones (>5 %, purple to blue). On the 1997 hazard map, we note that the longest flows that reached the coast and the 2007 flow field was emplaced in low probability zones (<1 %; yellow–green). Because our hazard maps are computed with a database in which only four of the 137 eruptions since 1931 are high volume, source-related-cycle terminating, events (i.e. 1931, 1961, 1986 and 2007), such infrequent events have a low probability and hence may occur in low probability areas. For example, it is clearly visible that the April 2007 lava flow occurred in a low (<0.5 %) probability zone of the 1997 hazard map (Figure 7). It is therefore important to recall that low probability does not mean that an event cannot happen, it only means that it is less probable, i.e., it is atypical if it happens in a low probability area. This is a major issue in terms of risk assessment because such atypical events have exceptionally high lava discharge rates (for example in 2007, discharge rates were sustained at more than 100 m$^3$/s over 30 days; Staudacher et al., 2009), with flows advancing rapidly to cut the belt road and enter the ocean within hours of the eruption onset (Harris and Villeneuve, 2018a). As a result, atypical events are capable of rapidly inundating large areas to cause extensive damage. They thus represent the highest risk effusive event at Piton de la Fournaise. Dedicated studies on the probability of occurrence of such high magnitude and intensity, but atypical, events need to be conducted, and a separate set of hazard maps are required to compute where and when such events are more likely to happen. Likewise, our analysis does not consider the poorly studied, but relatively recent (post-1708), long-lasting activity related to overflow from summit lava lakes, as was common between 1750 and 1800, and again around 1850 (Michon et al., 2013; Peltier et al., 2012). Our maps are, though, applicable to the most common effusive event scenario currently encountered at Piton de la Fournaise. However, they must be used and applied with the above caveats in mind regarding the type of activity and effusive event to which they apply.

## 4.2. Historical and geological records: representativeness of future eruptions

Our geographical distribution of vent density per unit area (Fig. 3b) is based on an inventory of 726 vents and includes pre-historic vents (scoria cones) identified in the geological record back to at least 57 ka (McDougall, 1971). The lava flow inventory and recurrence probability inside the Enclos covers a time span of 88 years (from 1931 to 2019), and outside the Enclos the record spans 311 years (from 1708 to 2019). Given the frequency of activity, this allows a comprehensive data set of 220 lava flows to be used (Table S1). These relatively long periods and large numbers of cases, mean that the hazard map of the entire volcano provides lava inundation probabilities relevant at a temporal scales spanning decades to a few centuries, and is based on a statistically robust data set. As shown in Figure 7, the lava inundation probability distribution may change over the time span of a few years. However, although precise hazard maps for such active centers need to be regularly updated for short term hazard assessment (Harris et al. 2019), the topography of the Enclos remains broadly the same (Figure 8), where the slope profile is downhill from west to east, guiding flows towards the coast. Therefore, the map presented here is reliable

and trustworthy for long term land use planning and management. This includes agricultural practices, urban planning, road network planning, assignment of protected areas, park-use planning, implementation of trail networks and installation of subterranean and above ground electric, sewage and gas networks; as well as targeting of potential zones where repair and replacement will be necessary (cf. Tsang et al., 2020).

Nonetheless, although, we argue that this hazard map is trustworthy in the long term, it may not be adapted for the short term, i.e., the next few days or months and over small spatial scales, i.e., hundreds of meters. For short-term hazard assessment, given the frequent resurfacing of the Enclos, the topography will locally change (Figure 8). Emplacement of new lava flow units will cause subsequent flows to advance down paths that are displaced 10s to 100s of meters over what would have been the case had the initial, pre-eruption, topography applied (Harris and Rowland, 2015). As shown in Harris et al. (2019), to accurately and precisely forecast exact lava flow paths in the short term, the topography needs to be revised after each eruption, and even during relatively long-lasting (weeks-to-months long) eruptions. Hazard maps appropriate for such timeframes and spatial scales could be produced with the same methodology given here but with updated source term data, i.e., a DEM that includes the new topography. Away from the local anomaly created by emplacement of a new lava flow field or cone system, though, the original DEM and hazard map will still apply.

### 4.3. Accounting for spatiotemporal volcanic activity patterns

Activity at Piton de la Fournaise follows cycles that evolve over time scales of a few years to a decade or so (Derrien, 2019; Got et al., 2013; Peltier et al., 2009; Roult et al., 2012; Vlastélic et al., 2018). This cyclic eruptive pattern will control the location, intensity and magnitude of eruptions, and hence also the vent location, as well as length and area of inundation of the associated lava flows. Eruptive fissures tend to be closer to the summit craters (i.e., proximal eruptions are more probable) at the start of a cycle than at the end (when distal eruptions are more likely to occur). Moreover, cycle durations will depend on the magma supply rate, where the lower the supply rate the longer the cycle (Derrien, 2019). Cycles also often terminate with a relatively long-lived, high volume, and high intensity effusive event (Coppola et al., 2017; Peltier et al., 2009). Therefore, depending on the length of cycle and where, temporally, in the cycle we are, both the geographical distribution of the probability of vent opening and the length of associated flow can vary. Specific hazard maps could therefore be built based on the cycles and the "current" position in a cycle in terms of the vent location, intensity of events and length of flow associated with the phase of the cycle underway.

Piton de la Fournaise also experiences longer-duration (20-30 year-long) cycles that are related to systematic evolution in the fertility of the magma source (Vlastélic et al., 2018). Phases of increasing mantle fertility are related to increasing eruption frequency and erupted volume, as the mantle porosity increases thus promoting melt extraction. Such increasing phases of activity ultimately culminate in exceptionally voluminous, eruptions with high effusion rates, as well as summit collapse or pit-crater formation (Vlastélic et al., 2018). At other volcanic centers such as Hawaii or Etna, lava flow length has been related to effusion rate and lava volume (Harris and Rowland, 2009; Malin, 1980; Pinkerton and Wilson, 1994;

Walker, 1973). At Piton de la Fournaise further work is needed to test and quantify whether there is a relationship between lava flow length, volume and effusion rate, and if this could be related to cyclicity (Derrien, 2019; Vlastélic et al., 2018) and topography (i.e., slope conditions). Such relations could be used to produce specific hazard maps depending on where, in terms of time, we are in such a longer-term source-related cycle or where, in terms of space, the likely vent opening locations are.

## 4.4 Uncertainties related to numerical modelling

The DOWNFLOW stochastic approach computes lava flow inundation area by summing $N$ lava flow paths for a given random vertical perturbation ($\Delta h$) added to the DEM. The calibration therefore requires setting of best fit values for these two main parameters. The parameter space of Figure 4 shows that the required number of runs ($N$) needed to achieve a good fit with the lava flow area ($\mu > 0.5$) must, in this case, be at-least 2 000–5 000. Following Tarquini and Favalli (2013), a safe choice for $N$ to ensure statistically robust simulations and ensure model (and output map) robustness is 10 000 in all cases. The calibration for $\Delta h$ gives different results for pre- and post-2007 cases, being 5 m and 2 m, respectively. This has two explanations. The first is simply due to the difference in the DEM spatial resolution. The resolution is 25 m for the 1997 DEM and 5 m for the other two DEMs. The bigger is the pixel size, the higher $\Delta h$ is expected to be. The second explanation resides in differences in lava flow dimension characteristics. According to Favalli et al. (2005), $\Delta h$ represents the characteristic vertical height of an obstacle that the flow can overcome. This implies that, since 2007, lava flow dimensions have changed: they must have become thinner as they can now only overcome an obstacle if that obstacle is 6 m lower than prior to 2007. Indeed, the lava flows in our database are more voluminous, longer and thicker between 1997 and 2007 (mean length: 3937 m, mean volume: $17.6 \times 10^6$ m$^3$, mean thickness 6.4 m) than between 2008 and 2019 (mean length: 2543 m, mean volume: $6.0 \times 10^6$ m$^3$, mean thickness 4.4 m). This difference can be related to the cyclic activity. The period between 1997 and 2007 corresponds to the end of a "deep source" cycle (Cycle 3 of Vlastélic et al., 2018). As a result, lavas were increasingly associated with higher magma fertility and magma supply rates, to produce more voluminous flows at relatively high effusion rates. Hence, if lava flows are longer and thicker, they are able to overcome higher obstacles. However, the period after 2007 relates to the beginning of a new "deep source" cycle (Cycle 4 of Vlastélic et al., 2018), with lower fertility and output rates. During this period the reverse is true: flows were less voluminous, and hence shorter and thinner and able to only overcome lower obstacles. The calibration parameters used for DOWNLOW, especially $\Delta h$, thus must be determined for, and then applied to, a specific period of activity if the results are to be valid; and changed should there be an evolution in a cycle or eruption style.

## 4.5 Use for hazard mitigation at Piton de la Fournaise

In terms of hazard mitigation at Piton de la Fournaise, our maps are intended to provide information of value in planning actions implemented by local authorities and in building resilience to future eruptions. In doing this, we place a focus on protection of public infrastructure. Figure 6 shows the locations of all roads and hiking trails that are the most likely to be covered by future lava flows, and the lengths of each that may need to be replaced.

For the island belt road, we show that 4.5 km of the 9.5 km that crosses the Enclos is in an intermediate probability (>0.5 %) zone in terms of lava inundation (Fig. 6). We also show that even if the probability is low, the probability of lava flow invasion is not negligible for several municipalities located beyond the Enclos on the northeast and southeast flanks of the volcano (i.e., Saint-Phillipe and Sainte-Rose, Fig. 6). In terms of hiking trails, although the main trail to the summit crater is in a low probability zone, it has a very high probability of future vent opening (>2 %/km$^2$, Fig. 3). This is because when we compute the hazard map, any pixel close to the summit (at high altitude) has a contributing area (possible vent location from which the lava path would reach the pixel in question) that is much smaller (by up to 1000 times) than for a pixel that is at lower altitude. Sometimes, trails inside the Enclos Fouqué have been covered by lava, as for example in July 2018 when 400 m of trail in the Enclos was buried, and in February 2019 when 150-200 m of trail very close to the viewing platform for the summit crater was buried (OVPF reports, ISSN 2610-5101).

In the case of an imminent eruption, our hazard map will thus complement the real time information that OVPF shares with civil protection, who need to identify potential locations from where people may need to be evacuated and road sections that need to be closed in the event of an eruption (Peltier et al., 2020). In addition, land use plans (referred as *Plan de Prévention des Risques* in French law) at La Réunion do not currently take into account volcanic hazard. The presented map is thus also intended to aid and guide stakeholders in developing effective mitigation and land use plans that also take into account the main volcanic hazard, with the caveat that our maps are for a "typical" effusive event. Separate maps will need to be drafted in the future for atypical events, including high intensity events (such as distal and *Hors Enclos* eruptions), as well as for prolonged lava lake overflow at the summit. Additionally, the map should be updated whenever topography changes and a new DEM becomes available. However, we have here set-up a flexible, GIS-based, methodology that allows for ease of update. As a consequence, for a short-term or atypical scenario, this framework allows updating of the input data (DEMs, vent locations, flow properties) to quickly produce a probabilistic map or specific flow forecast scenario as and when needed.

## 5 Conclusion

Piton de la Fournaise is one of the most active effusive centers in the world, it being a volcano that has experienced more than one eruption a year since human settlement of the island at the end of the 17$^{th}$ century, and two eruptions a year since 1979 when the volcano began being monitored by OVPF. The resulting database available for calculation of lava flow frequency and probability, as well as future vent opening, thus comprises more than two hundred individual lava flows emplaced within the Enclos since 1931, about fifteen lava flows emplaced outside of the Enclos since 1708 and more than seven hundred vents over the entire edifice. Within the most active area of the volcano (i.e, the Enclos), a lava flow has been emitted on average every five months since 1931, thirteen of which have cut the island belt road, while beyond the caldera, an effusive event has occurred on average every twenty-one years over the last three centuries. Since 1931, two *Hors Enclos* eruptions (in 1977 and 1986) have caused property damage.

This large database thus allows compilation of a statistically robust hazard assessment and production of probabilistic hazard maps for lava flow inundation. Using this, we present the up-to-date lava flow hazard map for Piton de la Fournaise, with lava flow path projections based on the stochastic model DOWNFLOW, to identify those areas that are most likely to be impacted by lava flows during any future eruption, and to quantify the probability of inundation in the medium to long term (decades). The availability of three DEM built in 1997, 2010 and 2016 allows us to produce a series of hazard maps that evolve with time and topography. Fundamentally we stress the need for frequent update of DEMs at such frequently active volcanoes, where topography is constantly evolving to influence flow path. This is of particular importance for short term lava flow hazard assessment during ongoing eruptions where topographies change over time scales spanning hours to days, subsequently emitted flows having paths influenced by the presence of previously emitted flows. In addition, given that the volcanic activity follows cycles that cause the location, erupted lava volume, effusion rate and flow dimensions to evolve with time, both the maps and the parameters used for modelling need to be applied on a cycle-dependent basis. We here produce a map that applies to the frequent typical effusive events, where the less frequent high-volume events actually characterize the lowest probability areas. Dedicated studies for probability occurrence of such high-volume events thus need to be conducted to complete the hazard assessment. Our lava flow hazard map production methodology is, though, intended as a flexible approach that can be applied to frequently active effusive centers, producing up-to-date maps based on a continually evolving data base that can be updated as eruption conditions evolve. This is essential support for informed land-use planning, as well as for use in crisis response and in drafting of hazard mitigation, emergency management and disaster management plans (cf. Coppola, 2015).

**Supplementary material**

Table S1: Table of the inventory of eruptions and lava flows since 1708

Figure S1: Map of the lava flow axis measurements used to extract the lava flow length

Figure S2: Figure of the lava flows for each period of time (1931-1966; 1972-1992; 1998-2007; 2008-2019)

**Data availability**

All data shown in this study including the eruption inventory and GIS layers (georeferenced .shp files of the vents, fissures and lava flows as well as the calculated density function of vent opening and hazard maps) are freely available upon request at the OVPF or in the supplementary material.

**Authors contribution**

MOC lead this study, wrote the manuscript and made all figures and tables. MF executed the vent probability distribution, the DOWNFLOW calibration and computed the hazard maps; NV wrote the DEM section; AP, ADM and NV contributed to the

eruption inventory; AD and NV draw the lava flow outlines; PB compiled the GIS data. All authors contributed to implementing the discussion and to the writing of the article.

**Acknowledgements**

This research was fully funded by the Agence National de la Recherche through the project LAVA (Program: DS0902 2016;
Project: ANR-16 CE39-0009, http://www.agence-nationale-recherche.fr/ Projet-ANR-16-CE39-0009). This is ANR-LAVA contribution no. XX and Laboratory of Excellence ClerVolc contribution number XX. We thank Patrick Bachèlery for reviewing the eruption inventory and Simone Tarquini and Hannah Dietterich for their very constructive reviews that improved this article.

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

**Figures and Tables:**

**Figure 1:**

**a) La Réunion island (© Google Earth) where the green line delineates the two volcanoes: Piton des Neiges (PN) to the northwest and Piton de la Fournaise (PF) to the southeast.**

**b) General map of Piton de la Fournaise showing the eruptive fissures as mapped up to end of 2019 (blue lines) and the**
**buildings (in black) with the main municipalities, main roads (in brown - RN2 is the national road) and touristic trails (white lines). The main three rift zones: north 120° (N120), northeast (NERZ) and southeast (SERZ) and the two volcanic zone: Puy Raymond Volcanic Alignment (PRVA) and South Volcanic Zone (SVZ) are delineated by the dashed red lines as suggested in Bachèlery (1981) and Michon et al. (2015). The yellow contour outlines the Enclos depression that is separated in three regions by the dashed yellow lines: the Enclos Fouqué (EF), the steep slopes "Grandes Pentes"**
**(GP) and the eastern lower part "Grand-Brûlé" (GB). The dashed yellow line between EF and GP is situated at 1800-1700 m. a.s.l. while the one between GP and GB is around 500 m. a.s.l. The background is the hill-shade of LiDAR DEM from IGN – released in 2010 and coordinates are within system WGS84-UTM 40S. Buildings, roads and trails are from BD TOPO® IGN.**

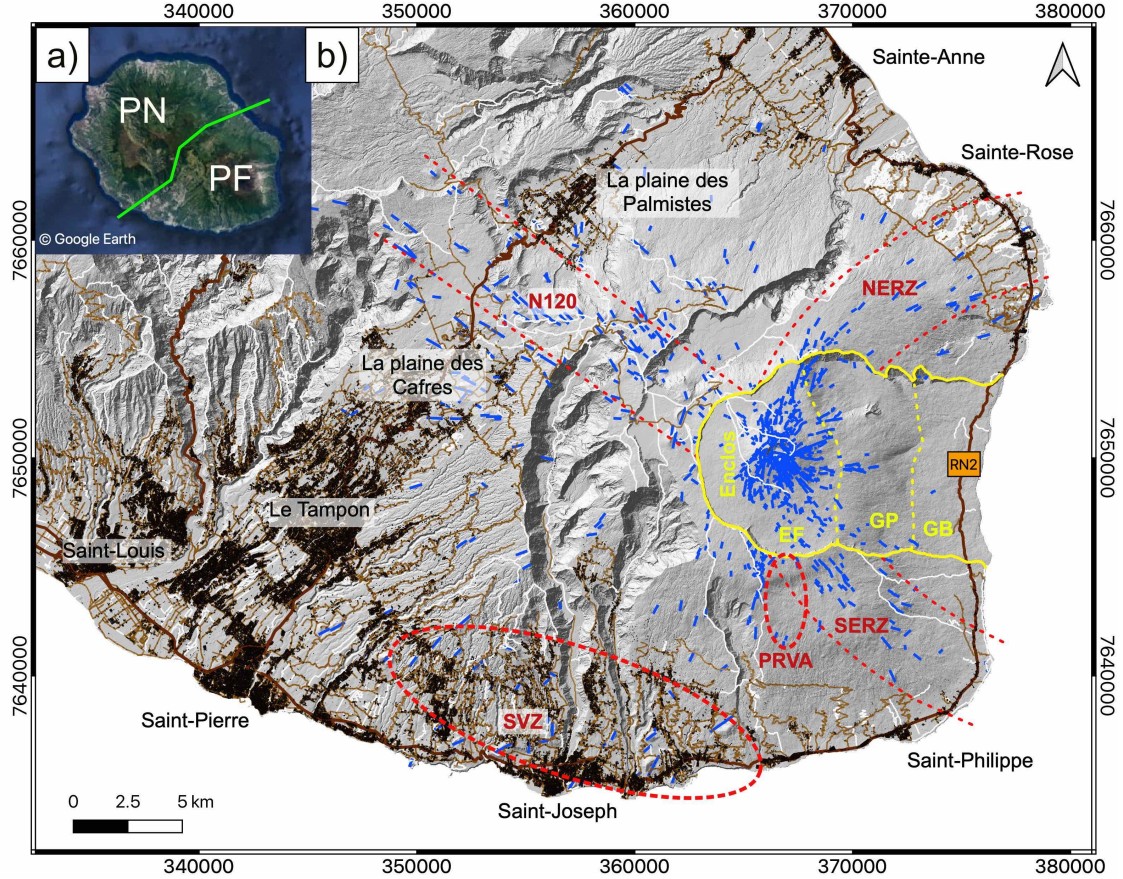

**Figure 2: Maps of the lava flows considered in this study.**

**a) Lava flows inside the Enclos since 1931 (date from which the lava flow mapping has been well recorded) up to end of 2019 from pale to dark red (see Derrien, 2019).**

**b) Lava flows outside the Enclos since 1708 and until end of 2019 including i) observed flows (pale orange to red) (see Michon et al. 2015): ii) not observed but mapped and dated lava flows (in blue and numbers indicate the year before BP for C14 dated flows – see Vergniolle and Bachèlery, 1982) and iii) tree-dated lava flows (these are not mapped but the datation location is represented by the green dots and associated numbers indicate the calendar year for flows - Albert et al. 2020). The yellow lines represent the limits between the regions we consider. The background is the hill-shade of LiDAR DEM from IGN – released in 2010.**

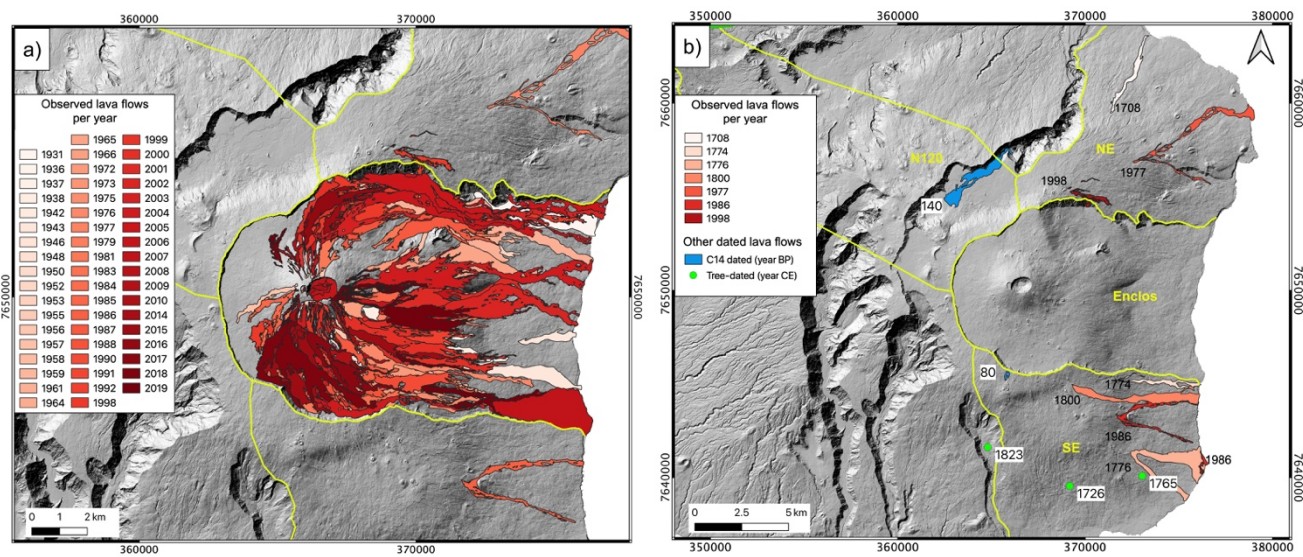

**Figure 3:**

a) Distribution and density (in number/km²) of scoria cones (black dots) on Piton de la Fournaise based on available inventory (OVPF database). In black are scoria cones older than 1931 and in white are the vents from 1931 to the end of 2019. In total they are 726 vents. The yellow lines represent the limits between the regions we considered to estimate probability of vent opening and compute the hazards maps: the Enclos, southeast (SE), northeast (NE), and north 120° (N120) rift zones.

b) Probability density function of vent opening (in %/km²). The dashed lines outline the rift zones to the N 120° (N120), to the northeast (NERZ) and to the southeast (SERZ) as well as the Puy Raymond volcanic alignment (PRVA) – (Bachèlery 1981; Michon et al. 2015).

The background is the hill-shade of LiDAR DEM from IGN – released in 2010.

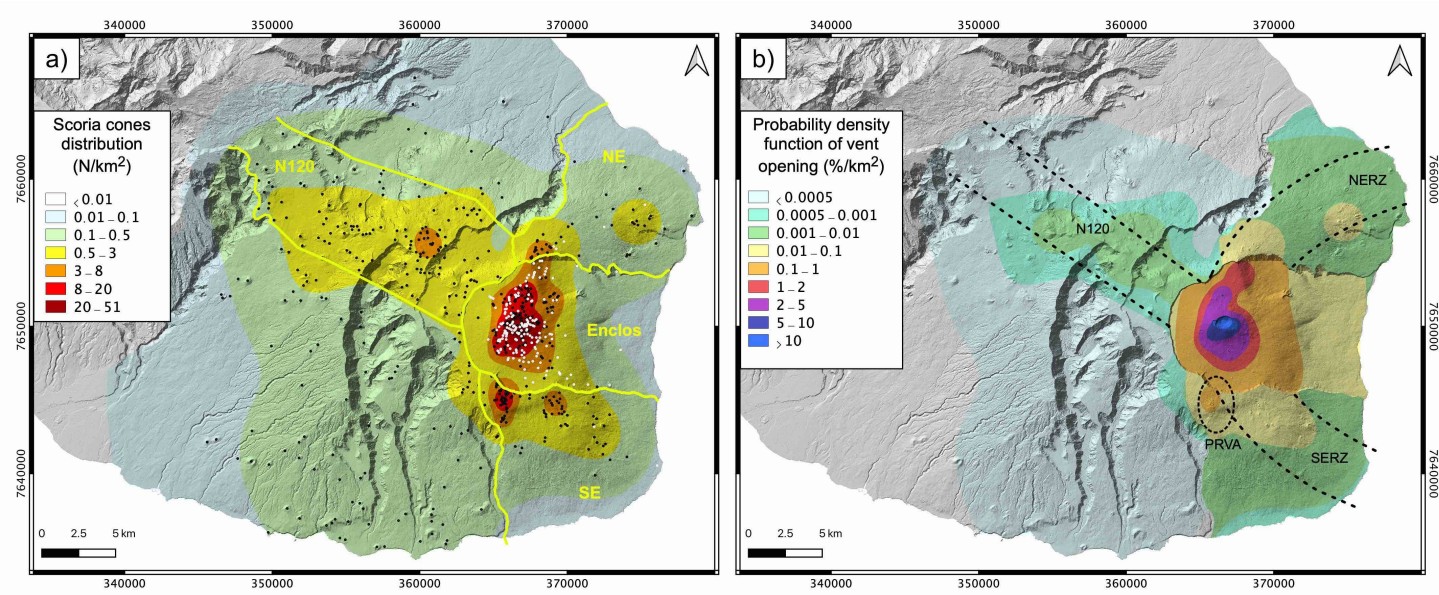

**Figure 4: DOWNFLOW calibration for a selected set of lava flows for the three time periods: a) on the 25 m resolution 1997 DEM; b) on the 2010 LiDAR DEM, c) on the 2016 DEM produced from Pleiades. To the left, the maps show the lava flow contours in white and the best DOWNFLOW simulations in blue. To the right, the best fit distribution over the *Δh - N* space (best-fit parameter range are in dark red and blue).**

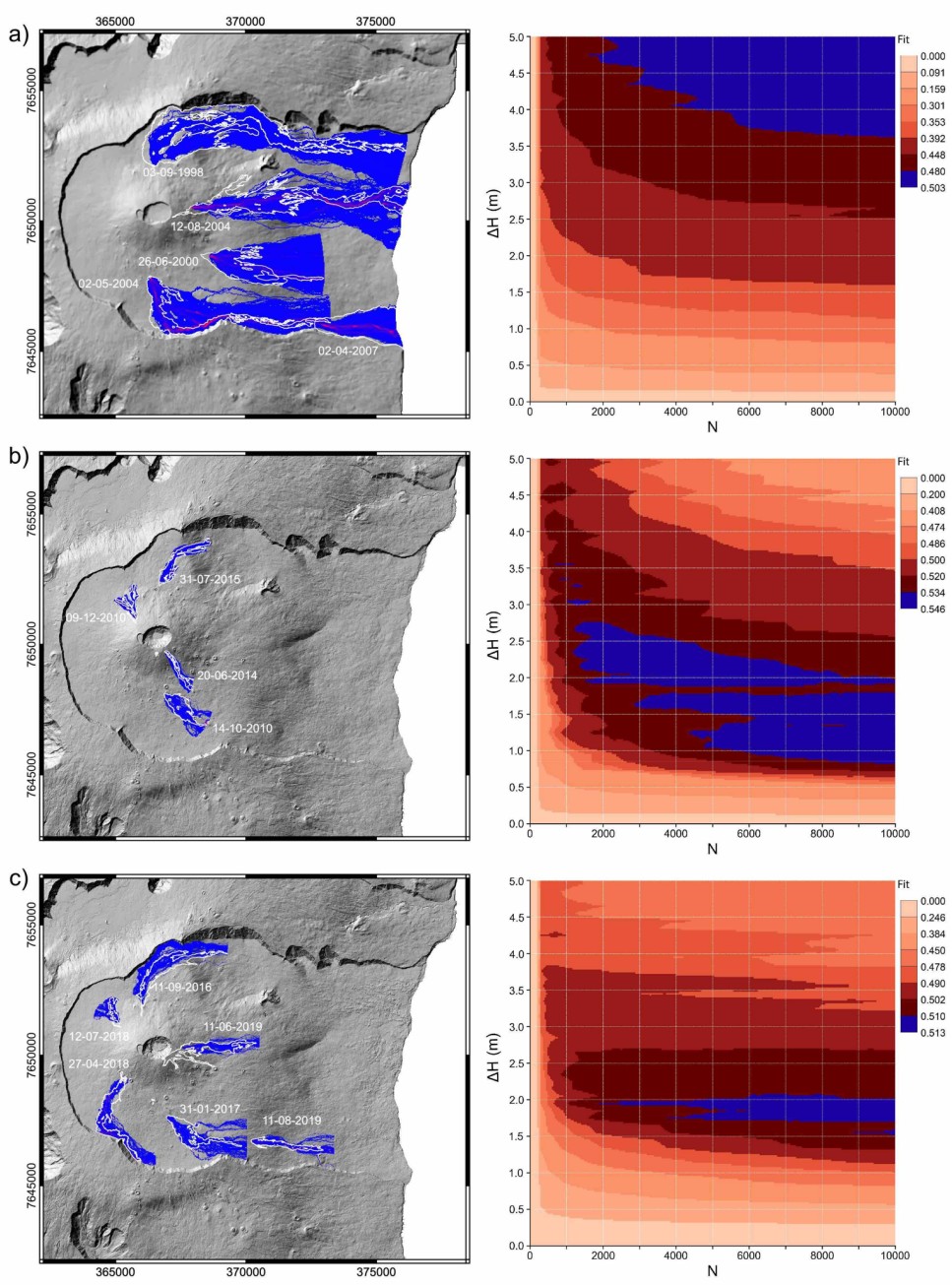

890

**Figure 5:  a) Frequency distribution of the lava flow length (by step of 500 m) at Piton de la Fournaise (La Réunion) for lava flows within the Enclos since 1931 and up to 1997, 2010, 2016 and 2019; and for all mapped flows outside the Enclos. b) Probability for a lava flow to reach a given distance (black line) and the corresponding lava flow length probability distribution (red line), for the 1931-2019 period, as function of distance from the vent.**

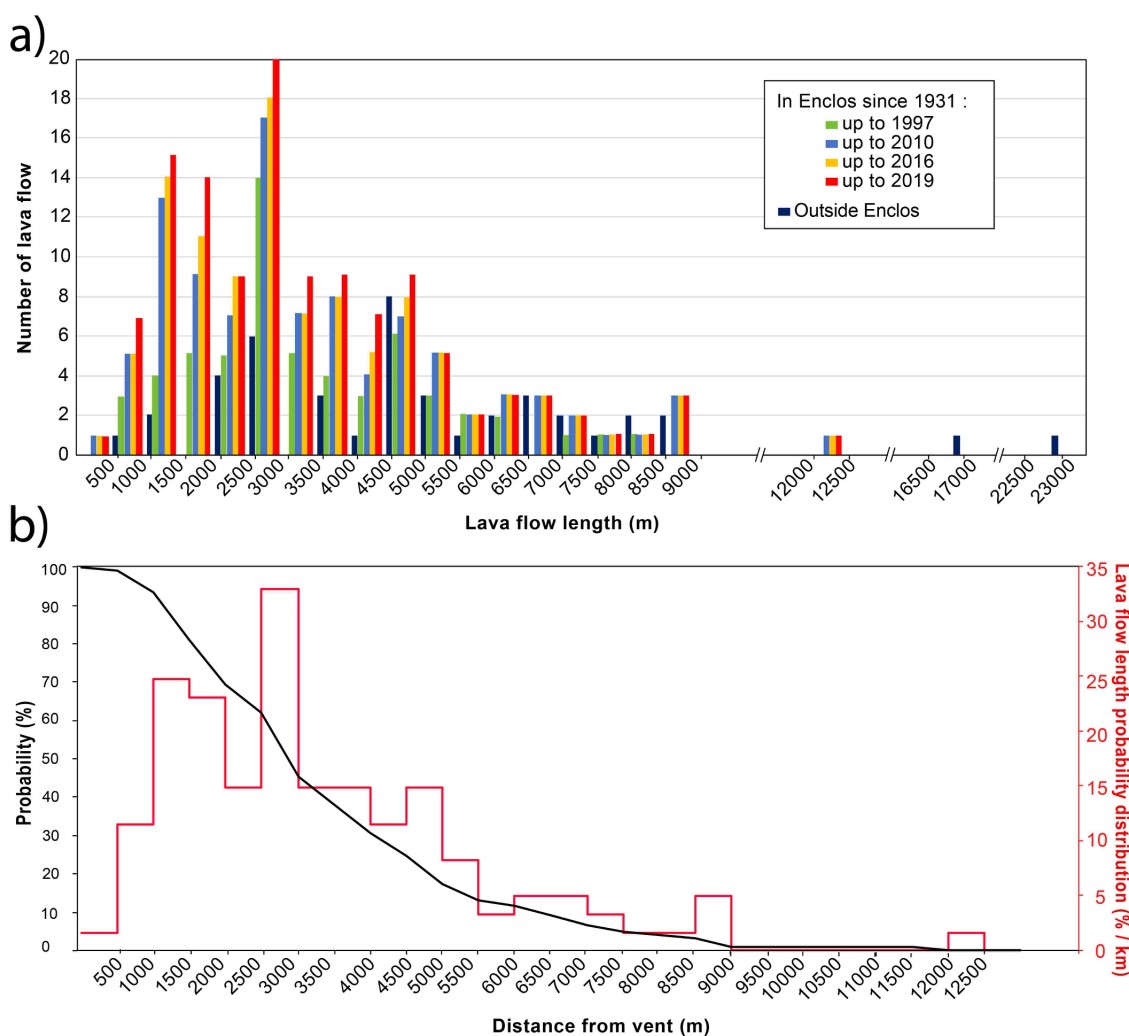

**Figure 6: Lava flow hazard map of Piton de la Fournaise. Probability of lava flow invasion is given as percentages and is color coded from extremely low (<0.01 %) to extremely high (>10 %). The buildings are represented as black polygons, and the main roads are in brown (in bold is the RN2 national road) and touristic trails are the white lines (data from from BD TOPO® IGN). The dashed red lines outline the rift zones to the N 120° (N120), to the northeast (NERZ) and to the southeast (SERZ) as well as the Puy Raymond volcanic alignment (PRVA) – (Bachèlery 1981; Michon et al. 2015). The dashed yellow lines separate the Enclos Fouqué caldera (EF) area from the steep slopes "Grandes Pentes" (GP) at 1800-1700 m. a.s.l. and the GP from the eastern lower part "Grand-Brûlé" (GB) at around 500 m. a.s.l. The background is the hill-shade of LiDAR DEM from IGN – released in 2010.**

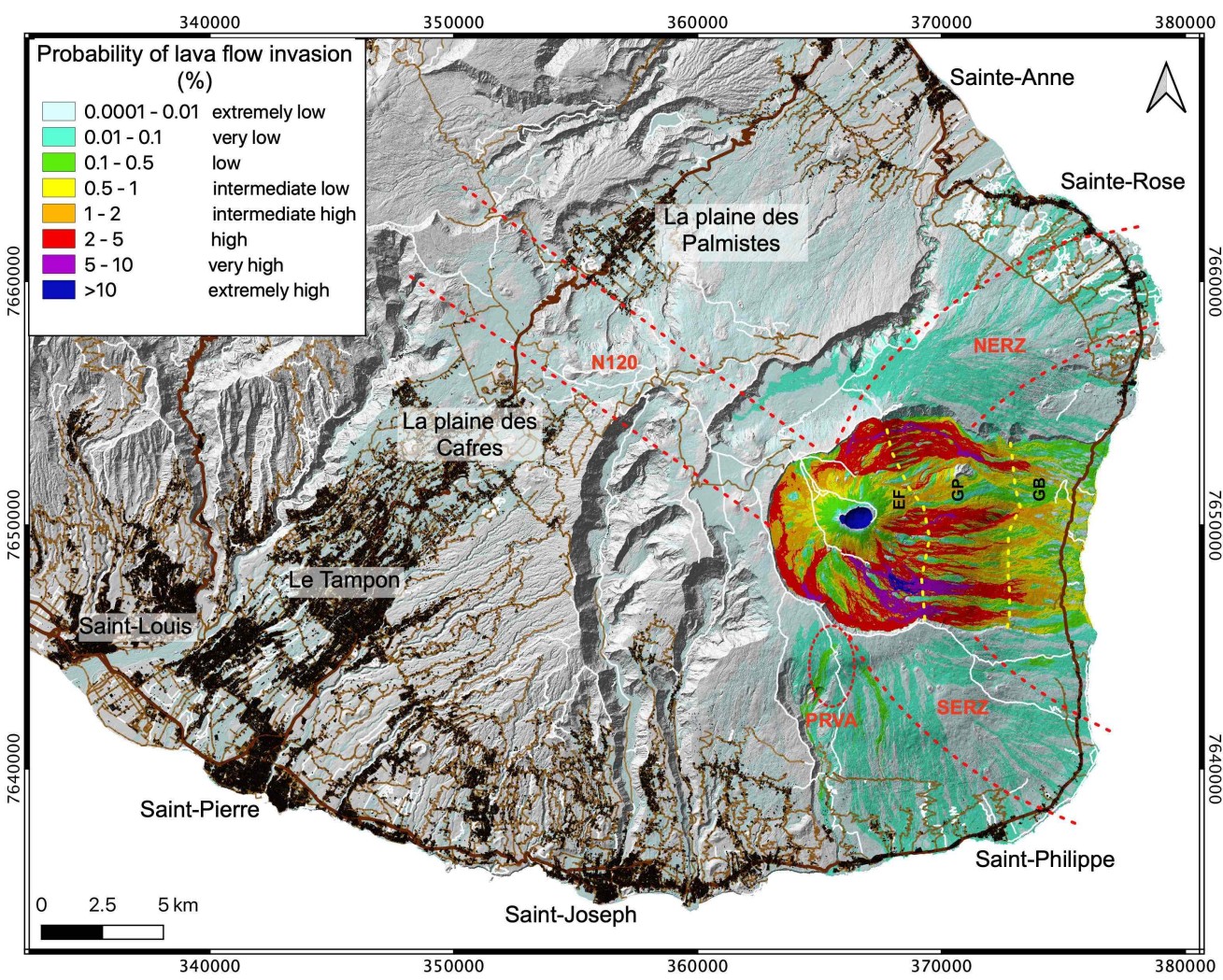

**Figure 7: Left column: Evolution of the lava flow hazard maps within the Enclos. The hazard maps are made on the DEM from 1997, 2010 and 2016, respectively, with the corresponding calibration (Fig. 4) and appropriated dataset since 1931 up to the corresponding date (Table 3, Fig. 5).**

**Middle column: successive annual lava flow coverage from the time of the DEM acquisition up to end of 2009 for the 1997 map and up to the end of 2019 for the 2010 and 2016 maps.**

**Right column: maps show the successive lava flows outline (as shown in the middle column) for the given time period.**

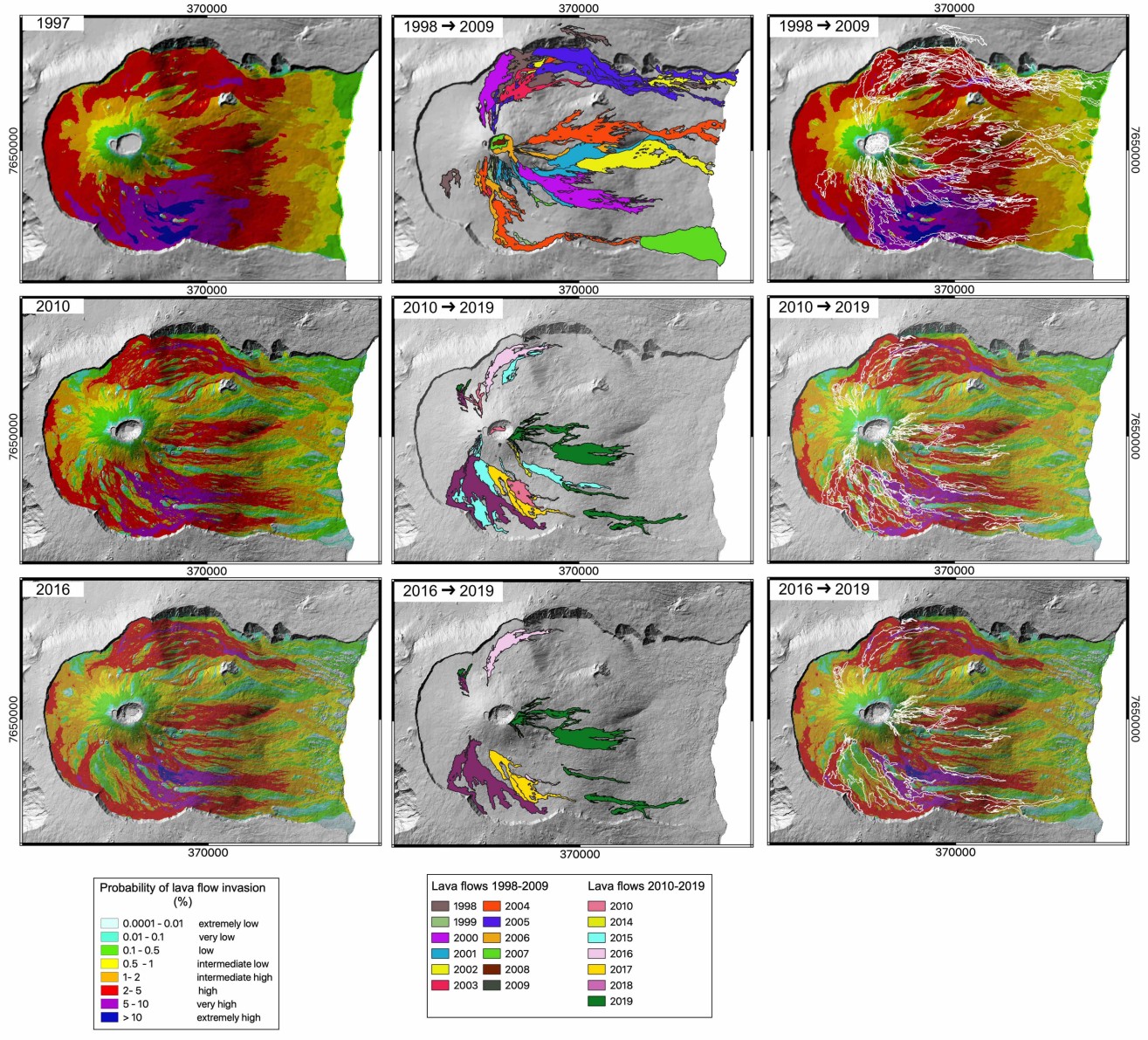

**Figure 8: Left column: difference in elevation between 1997 DEM and 2010 DEM (top) and between 2010 DEM and 2016 DEM (bottom). Note that 1) on the 2010-1997 map, the noise in differences is partly due to resolution difference and to slight mis-alignment, 2) on the 2016-2010 map, the positive difference near the cost is due to the vegetation (that has not been removed from the 2016 DEM). Right column: difference of the hazard maps (probability of lava flow invasion in %) built with the data up to 1997 and up to 2010 (top) and built with the data up to 2010 and up to 2016 (bottom).**

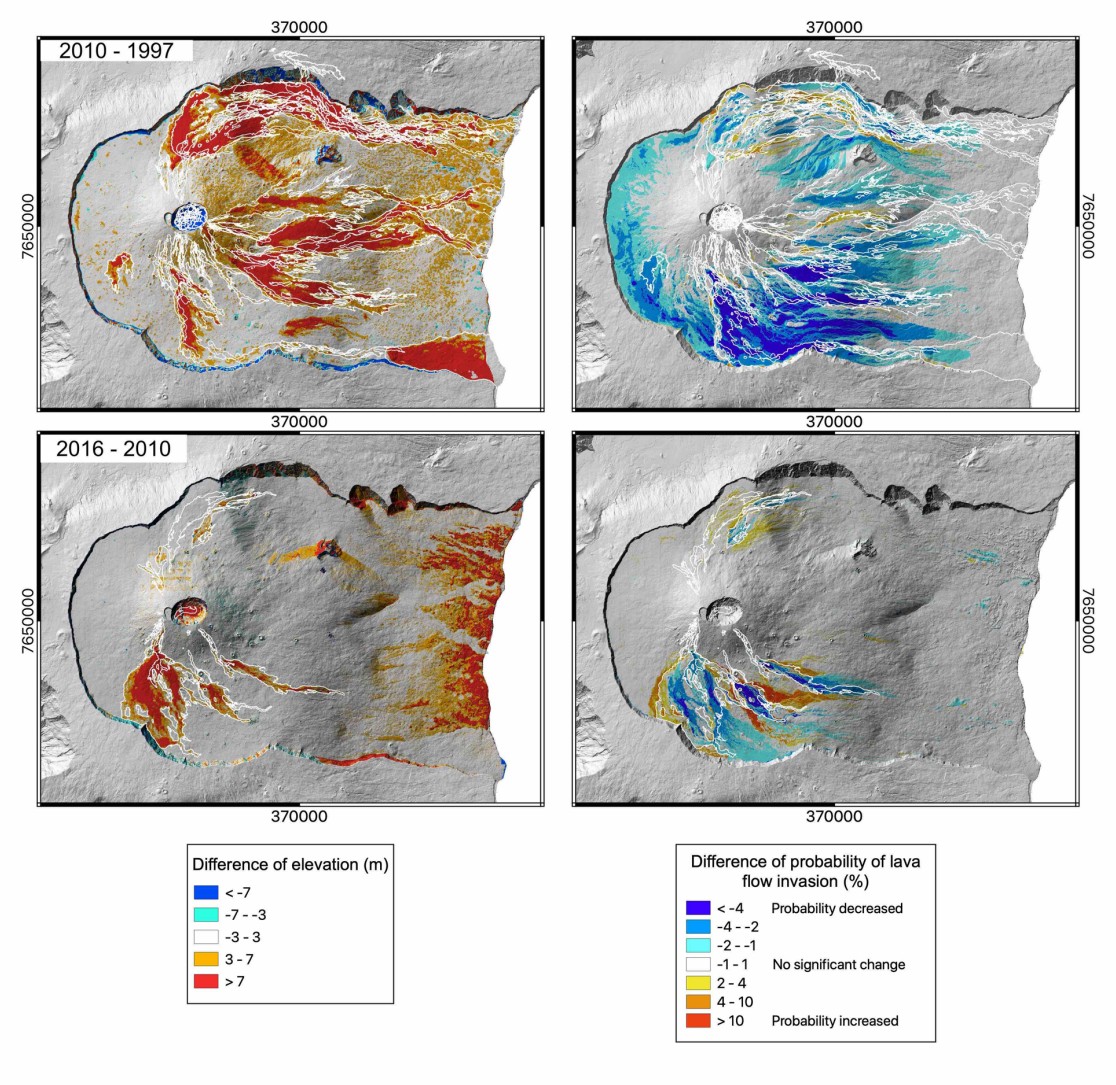

**Table 1: Number of lava flow and statistic of recurrence and relative occurrence probability (see Table S1 for inventory).**

| Region | N. lava flows | Time interval (years) | Recurrence time (year) | N. lava flows per year (year$^{-1}$) | Relative occurrence probability (%) |
|---|---|---|---|---|---|
| Summit craters | 47 | 88 | 1.9 | 0.53 | 23.8 |
| Enclos w/out craters* | 146 | 88 | 0.6 | 1.66 | 73.9 |
| *Inside Enclos*** | *193* | | *0.5 (5.5 months)* | *2.19* | *97.8* |
| SE | 9 | 290 | 32.2 | 0.03 | 1.4 |
| NE | 5 | 311 | 62.2 | 0.02 | 0.7 |
| N120 | 1 | 209 | 209.0 | 0.005 | 0.2 |
| *Hors Enclos *** | *15* | | *20.7* | *0.05* | *2.2* |
| Total | 208 | | | 2.2 | 100.0 |

*Enclos includes the Enclos Fouqué, Grandes Pentes and Grand-Brûlé, except the summit craters area

***Inside Enclos* includes the whole Enclos with the summit craters. Lava flows are counted since 1931 up to the end of 2019.

****Outside Enclos* lava flows are divided by region south of the caldera (SE), north (NE) and along the N120 rift zone (N120) (see Figure 2), [#] Time interval corresponds to the time between the oldest considered dated lava flow up to the end of 2019.

**Table 2: Scoria cone distribution divided by regions**

| Region | N. vents | Fraction of vents (%) |
|---|---|---|
| Summit craters | 27 | 3.7 |
| Enclos w/out craters* | 300 | 41.3 |
| *Inside Enclos*** | *327* | *45.0* |
| SE | 98 | 13.5 |
| NE | 49 | 6.7 |
| N120 | 142 | 19.6 |
| Rest of the volcano | 110 | 15.2 |
| *Hors Enclos**** | *399* | *55.0* |
| Total | 726 | 100 |

*Enclos includes the Enclos Fouqué, Grandes Pentes and Grand-Brûlé, except the summit craters area

***Inside Enclos* includes the whole Enclos with the summit craters. Lava flows are counted since 1931 up to the end of 2019.

****Outside Enclos* lava flows are divided by region south of the caldera (SE), north (NE) and along the N120 rift zone (N120) (see Figure 2),

**Table 3: Lava flow relative occurrence probability and scoria cone distribution within Enclos for the 1931-1997, 1931-2010, 1931-2016 and 1931-2019 periods**

| Region | N. lava flows | Time interval (years) | Recurrence time (year) | N. lava flows per year (year$^{-1}$) | Relative occurrence probability (%) | N. Scoria cones | Fraction of Scoria cones (%) |
|---|---|---|---|---|---|---|---|
| since 1931 up to 1997 | | | | | | | |
| Summit craters | 31 | 66 | 2.1 | 0.5 | 28.4 | 3 | 3.4 |
| Enclos w/out craters | 78 | 66 | 0.8 | 1.2 | 71.6 | 85 | 96.6 |
| Total | 109 | | | 1.7 | 100.0 | 88 | 100 |
| since 1931 up to 2010 | | | | | | | |
| Summit craters | 46 | 79 | 1.7 | 0.6 | 27.1 | 9 | 5.8 |
| Enclos w/out craters | 124 | 79 | 0.6 | 1.6 | 72.9 | 146 | 94.2 |
| Total | 170 | | | 2.2 | 100.0 | 155 | 100 |
| since 1931 up to 2016 | | | | | | | |
| Summit craters | 47 | 85 | 1.8 | 0.6 | 26.1 | 10 | 6.1 |
| Enclos w/out craters | 133 | 85 | 0.6 | 1.6 | 74.2 | 154 | 93.9 |
| Total | 180 | | | 2.1 | 100.00 | 164 | 100 |
| since 1931 up to 2019 | | | | | | | |
| Summit craters | 47 | 88 | 1.9 | 0.5 | 24.4 | 10 | 5.4 |
| Enclos w/out craters | 146 | 88 | 0.6 | 1.7 | 75.6 | 176 | 94.6 |
| Total | 193 | | | 2.2 | 100.0 | 186 | 100 |
