# Peer review of "Lava flow hazard map of Piton de la Fournaise volcano"

_Natural Hazards and Earth System Sciences, 2020_

## Referee Comment (RC1) · Simone Tarquini (Referee) · 29 Jan 2021

**General comments**

The manuscript illustrates a comprehensive work about the hazard posed by lava flows at Piton de la Fournaise volcano. The input data presented (DEMs, lava flow maps and relevant parameters among others) are certainly rich, and provide an ideal starting point to carry out the present analysis. As usual with this code, DOWNFLOW provides an incredibly rich database of simulations. In spite of the great databases, I think that the interpretation and discussion of the hazard maps (especially the time series for the enclos), should be slightly refined/adjusted. I also suggest to add some clarification in the use of the code. Finally, I suggest some refinements in the wording in places, where the editing has been probably hastily bundled. For sure, I have no severe concerns with the present work and I am sure it will be adequately refined.

**Major points (in order of line number)**

**Code tuning**  Lines 286-293 – here the authors describe the code calibration. Please remind to the readers which grid resolution are set for each DEM when they have been set up for simulations. This is not trivial, because the LIDAR DEM (in section 2) was described as having a mix of 5 and 1 m resolution. I can argue the authors used a 5 m grid, but clarify this. More importantly, if the authors used a 25 m cell size for the 1997 DEM and 5 m cell size for the other cases, could they explain whether or not such a strong difference has an impact in the output of the tuning? The random noise in elevation is applied at a rather different spatial frequency in the two cases (25 times more often per unit area for the 5 m DEM). In other probabilistic codes (I remember Q-Lavha, Mossoux et al 2016, for example), varying the cell size is part of the tuning session. I know this is not the case with DOWNFLOW, but I would appreciate if the authors can clarify this (possibly not negligible) point so as to avoid possible misinterpretations of results.

**Evolving haz map**  Lines 361-370 – This paragraph should be the summa of the whole work, but it is weak instead. The preceding sections painstakingly refined every single factor (vent opening pdf, length frequency, code calibration..), but all this ends up in a vague description. I think Figure 7 doesn't help enough (see suggested improvements). For example "we note the development of a high probability area to the north of the terminal shield" (lines 362-363). I am completely lost, unable to see it. Or "This is due to the high number of eruptions occurring to the north of the shield between 1998 and 2006" (lines 363-364); I do look at the rich data provided in Fig S2c, but it shows (to me) an essentially balanced number of eruptions to the north and to the south of the shield. More importantly, the above suggests that the authors are deeming a dominant factor the

modification of the pdf (new vents, then higher vent opening pdf, then higher hazard to the North). This could be a perfectly reasonable explanation, but the authors should note that this acts **against** the (often reminded) impact of morphological changes due to new lava deposits (e.g. "the evolution of the hazard maps resulting from changes in the DEMs" -line 82 in the introduction-). The local stacking of multiple lava flows to the North should act as a damper of the probability of future lava inundation to the North. The latter point is clearly shown by Tarquini and Favalli 2010 (see refs below) and Favalli et al 2011 at Mt Etna. However, Fig 7 (right column) shows that here flow fields tend to pile up multiple times in a few years. Is this an outcome of topographically confined areas? Otherwise, lava flows are so thin here, that they do not promote the usual topographic inversion (which inhibits superposition)? Both instances at once? I am not asking to clarify this – beyond the present scope – , but it is important to describe/clarify the different role played in the evolving haz map by factors such as changes in the vent opening pdf and DEM modifications due to new lavas.

For sure, an assessed step to strengthen the discussion about the evolving hazard in a clear and objective way, is to derive the maps of the hazard difference as shown in the references above.

To support statements such as "Figure 7 also illustrates that the hazard maps were able to predict where subsequent lava flows actually occurred" (lines 366-367) I suggest to overlap the contour of such subsequent lava flows to the haz map (as in Favalli et al 2009c and in the references above), otherwise the reader simply cannot grasp the (eventual) correctness of the statement.

The authors indicate the voluminous 2007 flow as a case of poor prediction of the map. They state the "issue" with this eruption is due to a vent at an exceptionally low elevation (i.e. in a low pdf area). The point of the eruptive cycles is clear and convincing (as discussed ahead), and exceptional end-cycle eruptions can be tricky. But I didn't find this point particularly intriguing.

I was intrigued, instead, by two evidences (if I see well enough) in the haz maps of Figure 7, which emerge when haz maps are compared with the following lava flows (right column Fig 7 or Fig S2):

1) –  In the 1997 haz map the purple (5-10%) and dark blue areas (>10%) form a broad peak covering almost completely the southernmost third of the EF. In contrast, the flows in Fig S2c appear piled up elsewhere (mostly <5%), with a poorly covered peak. Please check if I am right. It is possible that this suggests a "too high" peak of vent openings in the southern flank of the shield ?

2) – In the 2016 haz map (if I see correctly..) the 2018 lava flow covers, to a significant extent, a green (low probability) area midway between crater and southern rim of enclos. Being downhill from a peak of vent openings, this low haz area should be the result of a "topographic shield". It would be interesting to explore/explain why it has been nonetheless inundated (if this is true). If this comment turns out to be correct, it wouldn't "invalidate" the map, it would simply show that it is necessary to be cautious in using it.

**Additional specific comments (in order of line number)**

line 63 – "Bretar et al 2013" is this the correct reference? Check please

line 75 – modify "in a such this dynamic" into "in such a dynamic" (tentative suggestion..)

Line 81 – "from  three DEMs acquired"

line 140 – "open on the Grandes.."

Section 2

line 168 – perhaps removes all this ".., and indeed up-to-date, .."

line 169 – consider the deletion "For this study we could use  three DEMs"

line 170 – "vertical resolution" appears an odd wording, it could be "vertical accuracy" ? Please rephrase

lines 171-173 – please specify that this DEM is later referenced to (in Figs captions) as the "LIDAR DEM from IGN – released in 2010"

Section 5

If I am not wrong, the description of the DOWNFLOW code is slightly inaccurate. It is hard to suggest something about DOWNFLOW to the authors (being the code programmer among them..), but I feel uncertain about the current text and I have some suggestions

line 270 – I can propose to change "follow the steepest" with "follow approximately the steepest", so as to suggest why DOWNFLOW introduces the random perturbation to account for lava flow paths.

line 272-278 – I would say DOWNFLOW do not computes "all the possible flow paths" (line 273), but just a number N of steepest descent paths (which are later found to fit lava flow paths..); instead of a random noise "(with a value of $\pm\Delta h$)" a noise in elevation randomly varying within the interval $\pm\Delta h$ can be clearer?

Line 278 – "occurred immediately after" why the need of being "hurry"? it is important that the topography hasn't significantly changed in the meantime, but if the topography is (just locally) updated, the authors could have adequately tuned $\Delta h$ and N by considering not the first, but the second, or third.. or $100^{th}$ lava flow.

Line 280 – consider substituting "cut at actual the length" with "cut at the actual length" (but it could be an English wording I am unaware of..)

Line 284 – "If this ratio ($\mu$) is one then the two areas coincide perfectly and the simulation is valid", for sure, the simulation is not valid only if $\mu = 1$.. a similar wording do not help understanding, please reword.

I would ask the programmer of the code to check a bit closer this section.

Section 6

lines 304-305 – Please clarify the function for lava flow length frequency distribution. I've seen the histogram in Figure 5, and IF I correctly understand.. a step-wise, discontinuous function is straightforwardly derived from the plot of the number of flows in each bin obtaining normalized frequencies. It is possible to make this function a bit more explicit so as to clarify the function P(Lij) used in equation 2? I see that something is added more ahead at the end of section 7, but it is probably better to clarify upfront here (by adding also a further panel to Fig 5?)

Section 7

lines 311-312 – the array of computational vents, the different DEMs used and the number of simulations carried out is mixed in the same sentence in a somewhat confusing way, and the reader can make a picture about this only after reading the following sections. I would define at first the sole array of computational vents. If I correctly understand, the two portions of the array (inside and outside enclos) have been used for different purposes (different maps). Only the inside enclos array has been used multiple times over different DEMs (with different vent opening pdf and flow length etc.), while the outside enclos array is used only once, over the 2010 DEM. Perhaps, it could be also worthy to clarify that, in each haz map presented, only one DOWNFLOW simulation is considered for every single node of the array. This is just to make clear to the reader that – in contrast with what may happens with some other codes – a single DOWNFLOW simulation (combined with P(Lij)) is able to account for a a variety of scenarios.

Section 7.2

line 352 – "recurrence time" I am good with table 3 but I missed the use of a "time" parameter throughout the manuscript.. I remember that Favalli et al (2009c) included the "recurrence interval" (time) in a formula to quantify the hazard. Please make explicit how the recurrence time is used or that it is not used here.

Line 362 – "(above 1800 m asl)" please, at least in one of the figures (not in all figures, to avoid overloading information), includes elevation contour lines (e.g. 500 m spaced) to support the reading of the maps, otherwise people less familiar with the altimetry of the volcano can hardly follow similar hints.

Line 387 – "representativeness of future eruptions", future eruptions or past eruptions?

Lines 406-409 – "However, .. (Fig 6)." The clarity of this sentence could probably be improved, but I completely agree. All in all, after all the work presented (and pending upon further refinements..), it seems that the impact of the topographic changes is limited. It seems that the frequent topographic resurfacing due to new lava flows keeps a "smooth" topography here, and do not introduces new "cumbersome" features (e.g. thick lava bulges such as happens at Mt Etna) which substantially affect the paths of future flows.

Lines 410-445 – the point of the cycles (stressed on lines 426-427) could be confirmed/strengthened (or not) by plotting the length data of Figure 5 along a time abscissa. A tip could be to account for the bias obviously introduced by the length cut at the sea by considering the volume (Vlastelic et al 2018), thus using the volume data as a proxy for the missing full length (e.g. the 2007 flow length)

Section 8.1.2

as for the modeling, I would recommend to fix the point of the cell size (noted above) before stressing on the different $\Delta h$. I note that id you take $2\Delta h$ as the obstacle, then the obstacle height difference when $\Delta h = 2$ and $\Delta h = 5$ is 6 instead of 3 as reported (line 456). The comment about the difference between the post 1997 and post 2007 lava flows could benefit from the plot suggested above for lines 410-445.

9 Conclusions

Beside what stated, an idea could be to highlight the different weight of the topographic changes with respect to the paths of future lava flow (and thus hazard) at different volcanoes. At Piton this factor appears to have a mild effect, as opposed to what has been found at other volcanoes such as Etna. Recent lava flow maps derived for Nyamulagira (Smets et al. see below) suggest a behavior similar to the one observed in the enclos (frequent lava flow superposition suggesting limited topographic inversion).

Line 754 – ..(green dotes..)
Line 774 – remove reference to the green line (there is no inset as in Fig 1 here)

Figure 4 – left column, it is not easy to separate the inside and outside of the polygons of lava flows. What about a dotted pattern inside or some partial transparency?

Figure 6 remove "green line" from the caption

Figure 7 right column, I would find clearer if the intervals are set to 1998-2010, 2010-2016, 2016-2019.

Figure S1

In the legend, the black line is not the lava flow length, but rather a kind of "approximate lava flow axis" (which are then used to derive lava flow length).

Speaking only about the outside enclos lava flows (impossible to decipher the inside), it seems that there are several flows without an axis (e.g. towards the southern coast). Perhaps these have not been accounted for because of some reason I forgot now (please confirm in the caption if this is the case or resolve otherwise). Could you confirm that the long-stretched lava flow axis along the riviere the langevin – riviere the remparts are assesed as single lava flows (perhaps partly eroded or sunk below river deposits) ?

Figure S1 caption – perhaps substitute "Map of the extracted lava flow length" with something like "lava flow axes used to measure the maximum length of flows"

Figure S2 caption – the four intervals indicated in the current caption do not mach at all the labels.. and there are hiatuses (?). The present collection of lava coverage is extremely useful. In order to allow the reader to profit at best from this collection, I propose to increase the number of panels to further promote interpretation of haz maps.

**References (if not already cited in the manuscript)**

Mossoux, S., Saey, M., Bartolini, S., Poppe, S., Canters, F., & Kervyn, M. (2016). Q-LAVHA: A flexible GIS plugin to simulate lava flows. *Computers & Geosciences*, *97*, 98-109.

Smets, B., Wauthier, C., & d'Oreye, N. (2010). A new map of the lava flow field of Nyamulagira (DR Congo) from satellite imagery. *Journal of African Earth Sciences*, *58*(5), 778-786.

Smets, B., Kervyn, M., d'Oreye, N., & Kervyn, F. (2015). Spatio-temporal dynamics of eruptions in a youthful extensional setting: Insights from Nyamulagira Volcano (DR Congo), in the western branch of the East African Rift. *Earth-Science Reviews*, *150*, 305-328.

Tarquini, S., & Favalli, M. (2010). Changes of the susceptibility to lava flow invasion induced by morphological modifications of an active volcano: the case of Mount Etna, Italy. *Natural hazards*, *54*(2), 537-546.

---

## Referee Comment (RC2) · Hannah Dietterich (Referee) · 31 Mar 2021

**Lava flow hazard map of Piton de la Fournaise volcano**
Chevrel et al.

In this manuscript, Chevrel et al. present an eruptive history database and methodology to produce numerical lava flow simulations and a probabilistic assessment of lava flow inundation hazard at Piton de la Fournaise. Their method builds on previous work done by these authors at other volcanoes to incorporate the complex spatiotemporal eruptive history of Piton de la Fournaise and, as they importantly highlight, the impact of topographic change at a frequently eruptive volcano on evolution in lava flow hazard. Overall, the manuscript is very good and will be of interest to a broad audience. The data, methods, and analysis are all appropriate, but some elements of the methodology and key discussion points are missing or wanting of more details. Below I summarize some major comments, questions, and suggestions. Pardon the length, these comments are largely minor, this is just a topic I am very interested in. There are also a number of typos and word order errors that just require a close reading with fresh eyes by the authors to fix.

**Major comments**

*Using recent eruptions to assess methods and implications*
A unique and powerful element of this work is the assessment of lava flow hazard through time using DEMs and eruptive history data that evolve over recent decades. Although these are used to generally state that changing terrain impacts hazard mapping and that recent flows generally occurred in areas that were previously deemed more likely, a missed opportunity is to use these maps and data to do a full hindcasting assessment and validate the method being presented. How well does the map based on data up to 1997 work for eruptions after 1997? Can this be quantified (e.g., Bevilacqua et al. 2017)? Is it better initially and then gets worse as topography changes more and more? What about up to 2009? This study has the distinction of allowing this critical discussion of how well we expect maps like this to do, what timeframe they're useful for, and where they fail given the availability of data over time and very frequent eruptions. These are topics that are all touched on but they are not framed in this fundamental way, where a hindcasting assessment would organize and strengthen the key conclusions. Showing that the method works well for recent decades would demonstrate validity and support further applications. I recommend adding a section to the discussion on this ("Validating hazard mapping with recent eruptions") and accompanying Fig. 7 with quantification of how well the map did for later flows through time. It is also regularly mentioned, but never demonstrated how topographic changes impact hazard assessment. Since this is a major conclusion of this work, and testable with the time series of hazard maps and eruptions, this is a good opportunity to show the changes caused by topographic evolution with visuals.

*Treatment of spatiotemporal variability in eruptive behavior*
As mentioned and assessed throughout the paper, there is significant variability in the frequency and style of eruptions both spatially and temporally at Piton de la Fournaise, including rare rift zone events, frequent (but episodic?) summit activity, and cyclic patterns in

eruption location and magnitude in recent decades. Currently, these are integrated empirically into a conditional hazard map (probability of lava inundation in the next eruption) that incorporates historical data or geologic mapping at various timescales depending on location. However, in the discussion it is then stated that this approach does not integrate all magnitudes or timescales depending on what the "next" eruption is. Certainly, eruptive history that is left out of the input data for hazard mapping will not be represented (e.g., lava lake activity that is not well preserved in the geologic or historical records), but it is unclear why the authors also do not feel that the map represents events that are included (e.g., recent large eruptions that end each eruptive cycle). If the input data are not representative, it would be helpful to offer suggestions for how they could be represented moving forward, or utilize a different methodology for the treatment of these spatiotemporal patterns. Similarly, if the results have limited utility to specific time periods or scenarios, incorporating those explicitly, such as producing different conditional probability maps for specific conditions or regimes (e.g., just the large events that occur only after smaller ones) or a map of probability of any lava inundation over a given time period (1 year, a decade, given that the recurrence rates are well characterized), would help in the discussion. Overall, it seems like the map in Fig. 6 is perhaps more representative of the current state of the volcano, and applicable on the scale of years to decades, than the text implies, because of the relatively complete recent record.

In terms of discussing the results, adding/reorganizing the discussion around the "Impacts of cyclicity (or spatiotemporal variability in eruptions more generally) on hazard assessment" instead of methods-based section headings (e.g., 8.1.2), would help focus discussion on a key topic that repeatedly is mentioned throughout the other sections. These cycles are introduced in the background section and talked about a lot in the discussion as changing in location and magnitude through time, but the reader is not shown what these look like or exactly how their properties evolve. This discussion should therefore be accompanied by a figure that shows this well. Maybe marking vent locations and flow extents by cycle position (in different colors) in map view and comparing this to hazard mapping results.

*Methods questions*
Input data for hazard mapping includes vent locations, as well as statistics on the number of lava flows, their lengths, etc. However, these are not clear in the methods given the potential for a given eruption to produce vents that are actually lengthy fissures, multiple vents, and multiple lava flows. Some clarification on how vents, flows, and locations were defined, any implications of these choices on the results, and references to literature on these challenges (e.g., Cappello et al. 2012; Runge et al. 2014), are needed. Given some uncertainty in how these may be defined, the high precision and number of significant figures in the tables seems overestimated and should be discussed.

For the application of DOWNFLOW, were these simulations run from a single vent or from all vents and/or a fissure geometry (lots of point sources along a line, say)? How were ocean-limited flows treated in defining the lava flow length distribution (line 305)? The DEM resolution is very important to the performance of DOWNFLOW, and separate calibrations are appropriate, so please provide the resolutions when describing the methods (e.g., line 287) and

in the discussion of the results (line 453). In Fig. 4a, it is not clear that the calibration was completed though, given the best-fit seems unconstrained in the parameter-space ($\Delta h > 5$?). How do multiple vents affect DOWNFLOW modeling/misfit? For long flows, the fit seems much worse (Fig. 4a) - greatly overpredicting the inundated area. Would a different model work better for these? Were other models considered? Can uncertainty from this type of misfit be propagated in this methodology?

Additionally, the impact of DEM resolution is worth testing with regard to varying $\Delta h$ between time periods/DEMs, rather than attributing all differences to thickness changes (e.g., 455). Coarser DEMs (such as the 1997 DEM v. the >2008 DEMs) have built-in flow spreading from pixel size, potentially greater DEM uncertainty, and significant effective smoothing, which generally seems to change the best-fit $\Delta h$ significantly. If you resample the later DEMs to 25 m instead of 5 m, does the best-fit $\Delta h$ change? There may be changes related to flow thickness, too, but DEM resolution should be integrated into this discussion (also in terms of validation – does the 25 m data perform worse?).

*Quantitative analysis of hazard probabilities*
There are a number of places where results are described, but using inundation values for a given pixel, rather than integrating the results to answer the broader questions, such as "what is the probability of lava inundation in the Enclos during the next eruption?" (line 22) or "what is the probability that the next lava flow will intersect the coastal road?" (line 339, this is written as if integrated, but it's just the probability within a given pixel, not along the whole road). The actual probabilities are possible to calculate with these data though (the first is dictated empirically, but the second could be calculated based on the model results).

**Line-by-line comments**

33 – Many lava flow hazard maps also incorporate time (probability of inundation at a location over a given time interval), as opposed to just a conditional probability on the condition of an eruption occurring (e.g., Bebbington 2013; Cappello et al. 2015). Given the known recurrence rates in this study, this extension would potentially be possible here as well, and relevant for applications to land use planning.
64 – "complete" – that seems risky statement given how extensive burial is also described
86 – "Morphological" – replace with "Geological"
120 – Introducing these here as low frequency, high impact events will set up why they are so difficult to forecast in the discussion. They're in the data, but they are not the "most likely" event, and less spatially concentrated.
150 – Expand this to fully introduce the spatiotemporal patterns in eruptive history. Given that episodes on the order of centuries are also later invoked for the methods of database assembly and the short/long-term applicability of the analysis, these could also be introduced here in this background section.
192 – Using the longest flows only will overpredict flow length and yield higher probabilities overall. Perhaps add more explanation (as mentioned in the 'methods' questions above)

Tables 1 and 2 – Are all flows mapped or are some flows that are unmapped (or poorly mapped) but were reported included in these counts? I think since these are directly compared to each other and they have the same rows, it would be much easier for the reader to just combine them into one table.

236 - Is this every single cone, spatter rampart, and fissure? Or grouped/summarized somehow?

242 – What function is used for bandwidth, is it symmetric or asymmetric (or fissure geometries) and why?

Section 7 – The first sentence sets up the goal of 'the next effusive eruption", but this is also making retrospective maps for other timespans, so make sure the plan of making multiple hazard maps that represent different times to look at any changes over time is introduced at the start of section 7. (will make line 323 less confusing on the first read).

Sections 7.1 and 7.2 are interpretive and should be in the discussion. 7.1 can become 8.1 and describe the results, including the integrated example probabilities mentioned above. 7.2 can be modified/replaced into the core validation exercise, comparing the maps and subsequent flows more quantitatively/explicitly to show how well the method works. Overlaying flow outlines directly on the hazard maps may be especially helpful. Where it works/doesn't work will then set up the following existing/re-organized discussions of spatiotemporal cyclicity, short/medium/long-term applications, importance of topographic updates, etc.

334 – Clearly state that these are conditional probabilities

Section 8.1.1 - Overall, a 'long-term' hazard map using present-day topography in a frequently active area is most useful on the scale of the next few decades. A short-term forecast map (hours to a month, say) should be using short-term vent opening probability map integrating monitoring data, not the whole volcano over centuries (and up-to-date topography). A century-scale map needs to incorporate the potential of the volcano to change regimes entirely (it's been stated that eruption frequency can vary spatially and temporally over centuries at Piton de la Fournaise) and the present-day topography is expected to change over time (although it will likely be similar for a long time, large-scale subsidence and such can happen in addition to lava emplacement). So maps like this are most useful in that next years-decades period mostly useful for annualized to lifetime-scale planning and risk, but not shorter or longer term. Products for these other timeframes can also be produced with the same methodology, just with different input data, and that is worth saying here.

425 – This contradicts line 303 (although it's broadly true of course!)

429–445 – It seems like these events are represented accurately in the input data (infrequent, less spatially confined), and thus the hazard map. That they occur very sporadically at locations that erupt infrequently does not mean the map is wrong (low-likelihood, large-magnitude). Based on this work, are there recommendations for how these should be treated, or maybe presented differently (e.g., as "worst-case" scenarios or similar)? What work would be needed to include lava lake activity in a future hazard map?

471 – A good place to include some numbers of the integrated probability of trails being inundated in the next eruption, or visually showing the exposure on the map.

477 - Can you reconcile this? It is an aspect of drainage/integration by accumulation from many possible vents, but it's extremely nonintuitive to readers and the public. Would it be valuable to

make a "proximal" hazards layer, say, using a vent opening contour to define a proximal hazard region subject to tephra, ballistics, gasses and near-vent lava flows?

487 - Explicitly highlight that in a short-term or atypical scenario, this framework allows updating of the input data (DEMs, vent locations, flow properties) to quickly produce a probabilistic map or specific flow forecast scenario as needed.

Fig. 6 – I don't see green or white lines. Perhaps adjust line widths and colors?

---

## Author Comment (AC1) · 1 Jun 2021

**RESPONSE TO REVIEWS**
Reviewer comments are in black and *our answers are in blue*

*Dear Editors and reviewers,*

*We acknowledge these meticulous reviews as they improved the manuscript organization and clarify many aspects of the methods as well as the results. Also, they allowed us to deepen the discussion by pointing out the limitations and the strengths of our approach, as well as our crucial interpretation of hazard maps.*

*All major comments were answered and the text improved as required. These changes include new numbering of the sections by gathering methods and data and providing a clearer results section and a longer and deeper discussion. Figure 5 has been improved. Figure 7 has been improved as recommended by both the reviewers and is now described in more details in the results section and well discussed in the discussion section. Anew figure (figure 8) was added to support discussion. All minor corrections were done.*

**REVIEWER 1 : SIMONE TARQUINI**

**General comments**
The manuscript illustrates a comprehensive work about the hazard posed by lava flows at Piton de la Fournaise volcano. The input data presented (DEMs, lava flow maps and relevant parameters among others) are certainly rich, and provide an ideal starting point to carry out the present analysis. As usual with this code, DOWNFLOW provides an incredibly rich database of simulations. In spite of the great databases, I think that the interpretation and discussion of the hazard maps (especially the time series for the Enclos), should be slightly refined/adjusted. I also suggest to add some clarification in the use of the code. Finally, I suggest some refinements in the wording in places, where the editing has been probably hastily bundled. For sure, I have no severe concerns with the present work and I am sure it will be adequately refined.
Major points (in order of line number)

Code tuning
Lines 286-293 – here the authors describe the code calibration. Please remind to the readers which grid resolution are set for each DEM when they have been set up for simulations. This is not trivial, because the LIDAR DEM (in section 2) was described as having a mix of 5 and 1 m resolution. I can argue the authors used a 5 m grid, but clarify this.
*This is now clarified (section 2.1)*

More importantly, if the authors used a 25 m cell size for the 1997 DEM and 5 m cell size for the other cases, could they explain whether or not such a strong difference has an impact in the output of the tuning?
*Yes, the DEM resolution has an impact on the calibration of the model. Although we did not test it here, it is actually well known. We added explicitly this information 'Note that the difference in DEM resolution (25 m for 1997 and 5 m for 2010 and 2016) implies that the random noise in elevation ($\Delta h$) is applied on a different spatial frequency. On a given*

*topography, the lower the pixel size the greater is the amount of random noise that needs to be applied."*

The random noise in elevation is applied at a rather different spatial frequency in the two cases (25 times more often per unit area for the 5 m DEM). In other probabilistic codes (I remember Q-Lavha, Mossoux et al 2016, for example), varying the cell size is part of the tuning session. I know this is not the case with DOWNFLOW, but I would appreciate if the authors can clarify this (possibly not negligible) point so as to avoid possible misinterpretations of results.

*Yes you are right this is not the case for DOWNFLOW and when using a 5m DEM we actually have 25 times more perturbation to apply than with a 25m.*
*At the end of section 5 we added:*
*"Note that the difference in DEM resolution (25 m for 1997 and 5 m for 2010 and 2016) implies that the random noise in elevation (Δh) is applied on a different spatial frequency (on a given topography, the lower the pixel size the more random noise is applied)."*

Evolving haz map Lines 361-370 – This paragraph should be the summa of the whole work, but it is weak instead. The preceding sections painstakingly refined every single factor (vent opening pdf, length frequency, code calibration..), but all this ends up in a vague description.
*This part of the manuscript was rewritten in order to comply with both reviewers' suggestions, now section 3.2*

I think Figure 7 doesn't help enough (see suggested improvements). For example "we note the development of a high probability area to the north of the terminal shield" (lines 362-363). I am completely lost, unable to see it. Or "This is due to the high number of eruptions occurring to the north of the shield between 1998 and 2006" (lines 363-364); I do look at the rich data provided in Fig S2c, but it shows (to me) an essentially balanced number of eruptions to the north and to the south of the shield.
*Figure 7 has been improved according to both reviewers' suggestions*

More importantly, the above suggests that the authors are deeming a dominant factor the modification of the pdf (new vents, then higher vent opening pdf, then higher hazard to the North). This could be a perfectly reasonable explanation, but the authors should note that this acts against the (often reminded) impact of morphological changes due to new lava deposits (e.g. "the evolution of the hazard maps resulting from changes in the DEMs" -line 82 in the introduction-). The local stacking of multiple lava flows to the North should act as a damper of the probability of future lava inundation to the North. The latter point is clearly shown by Tarquini and Favalli 2010 (see refs below) and Favalli et al 2011 at Mt Etna. However, Fig 7 (right column) shows that here flow fields tend to pile up multiple times in a few years. Is this an outcome of topographically confined areas?
*Indeed flows tend to pile up in the south of the terminal shield. As it can be seen in Figure 7, although the emplacement of the large 2015 eruption changed the topographies and therefore affect the probability of future lava flow invasion toward lesser value, this did not prevent the 2017 flow to emplace exactly there. This is now explain in section 3.2 and using figure 8.*

Otherwise, lava flows are so thin here, that they do not promote the usual topographic inversion (which inhibits superposition)?

Both instances at once? I am not asking to clarify this – beyond the present scope – , but it is important to describe/clarify the different role played in the evolving haz map by factors such as changes in the vent opening pdf and DEM modifications due to new lavas. For sure, an assessed step to strengthen the discussion about the evolving hazard in a clear and objective way, is to derive the maps of the hazard difference as shown in the references above.

*We accordingly clarify this point (section 3.2 and using figure 8)*

To support statements such as "Figure 7 also illustrates that the hazard maps were able to predict where subsequent lava flows actually occurred" (lines 366-367) I suggest to overlap the contour of such subsequent lava flows to the haz map (as in Favalli et al 2009c and in the references above), otherwise the reader simply cannot grasp the (eventual) correctness of the statement.

*Done*

The authors indicate the voluminous 2007 flow as a case of poor prediction of the map. They state the "issue" with this eruption is due to a vent at an exceptionally low elevation (i.e. in a low pdf area). The point of the eruptive cycles is clear and convincing (as discussed ahead), and exceptional end-cycle eruptions can be tricky. But I didn't find this point particularly intriguing.

I was intrigued, instead, by two evidences (if I see well enough) in the haz maps of Figure 7, which emerge when haz maps are compared with the following lava flows (right column Fig 7 or Fig S2): 1) – In the 1997 haz map the purple (5-10%) and dark blue areas (>10%) form a broad peak covering almost completely the southernmost third of the EF. In contrast, the flows in Fig S2c appear piled up elsewhere (mostly <5%), with a poorly covered peak. Please check if I am right. It is possible that this suggests a "too high" peak of vent openings in the southern flank of the shield ? 2) – In the 2016 haz map (if I see correctly..) the 2018 lava flow covers, to a significant extent, a green (low probability) area midway between crater and southern rim of enclos. Being downhill from a peak of vent openings, this low haz area should be the result of a "topographic shield". It would be interesting to explore/explain why it has been nonetheless inundated (if this is true). If this comment turns out to be correct, it wouldn't "invalidate" the map, it would simply show that it is necessary to be cautious in using it.

*Yes you saw correctly. This is now clarified in the discussion in the section "Validating hazard mapping with recent eruptions"*

**Additional specific comments (in order of line number)**

line 63 – "Bretar et al 2013" is this the correct reference? Check please

*Yes it is correct.*

line 75 – modify "in a such this dynamic" into "in such a dynamic" (tentative suggestion..)

*Done*

Line 81 – "from the three DEMs acquired"

*Done*

line 140 – "open ion the Grandes.."

*Done*

Section                                                                                    2

line 168 – perhaps removes all this ".., and indeed up-to-date, .."

*Done*

line 169 – consider the deletion "For this study we could use to three DEMs"
*Done*

line 170 – "vertical resolution" appears an odd wording, it could be "vertical accuracy" ? Please rephrase
*Done*

lines 171-173 – please specify that this DEM is later referenced to (in Figs captions) as the "LIDAR DEM from IGN – released in 2010"
*Done*

Section 5

If I am not wrong, the description of the DOWNFLOW code is slightly inaccurate. It is hard to suggest something about DOWNFLOW to the authors (being the code programmer among them..), but I feel uncertain about the current text and I have some suggestions.
*All suggestions were taken into account.*

line 270 – I can propose to change "follow the steepest" with "follow approximately the steepest", so as to suggest why DOWNFLOW introduces the random perturbation to account for lava flow paths.
*Done*

line 272-278 – I would say DOWNFLOW do not computes "all the possible flow paths" (line 273), but just a number N of steepest descent paths (which are later found to fit lava flow paths..);
*Done*

instead of a random noise "(with a value of ±Δh)" a noise in elevation randomly varying within the interval±Δh can be clearer?
*Done. (see ection2.4)*

Line 278 – "occurred immediately after" why the need of being "hurry"? it is important that the topography hasn't significantly changed in the meantime, but if the topography is (just locally) updated, the authors could have adequately tuned Δh) and N by considering not the first, but the second, or third.. or 100$^{th}$ lava flow.
*This was rewritten accordingly*

Line 280 – consider substituting "cut at actual the length" with "cut at the actual length" (but it could be an English wording I am unaware of..)
*Done.*

Line 284 – "If this ratio (μ) is one then the two areas coincide perfectly and the simulation is valid",) for sure, the simulation is not valid only if μ = 1.. a similar wording do not help understanding, please reword.
*This is rewritten as follow: "Under this condition, μ is a measure of the "goodness of fit" between simulated and actual parameters, where if μ = 1 then the two areas coincide perfectly and if μ →0 then the simulation becomes increasingly unrealistic. Best fit parameters are*

*usually obtained for μ =0.5 (Tarquini and Favalli, 2011). Proietti et al. (2009) and Spataro et al. (2004) evolve this approach slightly by considering a fitting function of e_1=√ μ. This yields the same results, but gives numerical values closer to one."*

I would ask the programmer of the code to check a bit closer this section.

Section 6

lines 304-305 – Please clarify the function for lava flow length frequency distribution. I've seen the histogram in Figure 5, and IF I correctly understand.. a step-wise, discontinuous function is straightforwardly derived from the plot of the number of flows in each bin obtaining normalized frequencies. It is possible to make this function a bit more explicit so as to clarify the function P(Lij) used in equation 2? I see that something is added more ahead at the end of section 7, but it is probably better to clarify upfront here (by adding also a further panel to Fig 5?)

*Yes, you understand correctly. This is now clarified in section 2.5 and a new graph (in figure 5b) was added to show an example of the probability function.*

Section 7 *(now section 2.6)*

lines 311-312 – the array of computational vents, the different DEMs used and the number of simulations carried out is mixed in the same sentence in a somewhat confusing way, and the reader can make a picture about this only after reading the following sections. I would define at first the sole array of computational vents. If I correctly understand, the two portions of the array (inside and outside enclos) have been used for different purposes (different maps). Only the inside enclos array has been used multiple times over different DEMs (with different vent opening pdf and flow length etc.), while the outside Enclos array is used only once, over the 2010 DEM. Perhaps, it could be also worthy to clarify that, in each haz map presented, only one DOWNFLOW simulation is considered for every single node of the array. This is just to make clear to the reader that – in contrast with what may happens with some other codes – a single DOWNFLOW simulation (combined with P(Lij)) is able to account for a variety of scenarios.

*This part was re-written for clarification (see section 2.6)*

Section 7.2 *(now section 3)*

line 352 – "recurrence time" I am good with table 3 but I missed the use of a "time" parameter throughout the manuscript.. I remember that Favalli et al (2009c) included the "recurrence interval" (time) in a formula to quantify the hazard. Please make explicit how the recurrence time is used or that it is not used here.

*Here we use the recurrence time not to give a time constraint to the hazard map (as done in Favalli et al 2009c) but to correctly rank the overall probabilities of the future vent opening in the different areas since the spatial distribution of scoria cones fails to do so for the reason explained in section 2.6.*

Line 362 – "(above 1800 m asl)" please, at least in one of the figures (not in all figures, to avoid overloading information), includes elevation contour lines (e.g. 500 m spaced) to support the

reading of the maps, otherwise people less familiar with the altimetry of the volcano can hardly follow similar hints.

*The 1800 m asl outline is actually the dashed line between Enclos Fouqué et Grande Pentes in figure 1. The dashed lines were added to Figure 6 as well, and this is now specified in the figure captions.*

Line 387 – "representativeness of future eruptions", future eruptions or past eruptions?

*Future eruptions: here we want to discuss whether using the database we have (based on past eruptions) is good or not for future eruptions.*

Lines 406-409 – "However, .. (Fig 6)." The clarity of this sentence could probably be improved, but I completely agree. All in all, after all the work presented (and pending upon further refinements..), it seems that the impact of the topographic changes is limited. It seems that the frequent topographic resurfacing due to new lava flows keeps a "smooth" topography here, and do not introduces new "cumbersome" features (e.g. thick lava bulges such as happens at Mt Etna) which substantially affect the paths of future flows.

*It depends on the volume of lava extruded, in some cases the changes can have a dramatic effect. For example, in a published article (Harris et al. 2019) we show that the changes in topography due to the 2015 eruption (35.5 Mm$^3$) needed to be taken into account in order to model the trajectory of the April 2018 lava flows that occurred in the same area.*

Lines 410-445 – the point of the cycles (stressed on lines 426-427) could be confirmed/strengthened (or not) by plotting the length data of Figure 5 along a time abscissa. A tip could be to account for the bias obviously introduced by the length cut at the sea by considering the volume (Vlastelic et al 2018), thus using the volume data as a proxy for the missing full length (e.g. the 2007 flow length)

*Vlastelic et al 2018 have already plotted the volume against time to define the cycles. The length vs. time do not reveal something more than what was showed by Vlastelic et al. 2018. Unfortunately, the relationship between volume and length is not straightforward. We therefore have now added some nuances in our discussion.*
*See section 4.3*

Section 8.1.2 *(now section 4.3)*

as for the modeling, I would recommend to fix the point of the cell size (noted above) before stressing on the different Δh. I note that id you take 2Δh as the obstacle, then the obstacle height difference when Δh = 2 and Δh = 5 is 6 instead of 3 as reported (line 456).

*This is now corrected*

The comment about the difference between the post 1997 and post 2007 lava flows could benefit from the plot suggested above for lines 410-445.

*Done*

9 Conclusions *(now section 5)*

Beside what stated, an idea could be to highlight the different weight of the topographic changes with respect to the paths of future lava flow (and thus hazard) at different volcanoes. At Piton this factor appears to have a mild effect, as opposed to what has been found at other volcanoes such as Etna. Recent lava flow maps derived for Nyamulagira (Smets et al. see

below) suggest a behavior similar to the one observed in the enclos (frequent lava flow superposition suggesting limited topographic inversion).

Line 754 – ..(green dotes..)
*Done*

Line 774 – remove reference to the green line (there is no inset as in Fig 1 here)
*Done*

Figure 4 – left column, it is not easy to separate the inside and outside of the polygons of lava flows. What about a dotted pattern inside or some partial transparency?
*We found that it is actually better to outline the flows with an empty polygon rather than a dotter pattern or partial transparency. The outlines are now in white and more visible*

Figure 6 remove "green line" from the caption
*Done*

Figure 7 right column, I would find clearer if the intervals are set to 1998-2010, 2010-2016, 2016- 2019.
*The point here is also to show how good the hazard map would be if we only had done the hazard map in 1998, 2010 or in 2016.*

Figure S1

In the legend, the black line is not the lava flow length, but rather a kind of "approximate lava flow axis" (which are then used to derive lava flow length).
*Yes. This is now specified in the caption as well as in the legend*

Speaking only about the outside enclos lava flows (impossible to decipher the inside), it seems that there are several flows without an axis (e.g. towards the southern coast). Perhaps these have not been accounted for because of some reason I forgot now (please confirm in the caption if this is the case or resolve otherwise). Could you confirm that the long-stretched lava flow axis along the riviere the langevin – riviere the remparts are assesed as single lava flows (perhaps partly eroded or sunk below river deposits) ?
*Extracting the lava flow length outside the Enclos is difficult for the oldest flows because for example the flows in the southern flank were not entirely mapped, and it is therefore complicated to extract their length. These flows were therefore ignored (same for the outside Enclos area to the east of the Enclos). Therefore it is true that this close bias our results.*
*Both lava flow in the Riviere Langevin and Riviere des Remparts are considered as single long flow (although eroded).*

Figure S1 caption – perhaps substitute "Map of the extracted lava flow length" with something like "lava flow axes used to measure the maximum length of flows"
*Done as mentioned above*

Figure S2 caption – the four intervals indicated in the current caption do not mach at all the labels.. and there are hiatuses (?).

*Ups, indeed this was an error- corrected now*

The present collection of lava coverage is extremely useful. In order to allow the reader to profit at best from this collection, I propose to increase the number of panels to further promote interpretation of haz maps.

*This is the entire full collection of lava coverage as compiled by Derrien 2019 (provided in great details) and completed with the 2019 eruption.*

*This 4 panels represent the lava flows during the main cycles.*

*We prefer to keep them like this rather than increasing the number of panels. We add here that this collection is available upon request to OVPF-IPGP.*

*Improvement of Figure 7 has benefit from some of this lava flow coverage data.*

References (if not already cited in the manuscript)
*We included the references that were suitable*

Mossoux, S., Saey, M., Bartolini, S., Poppe, S., Canters, F., & Kervyn, M. (2016). Q-LAVHA: A flexible GIS plugin to simulate lava flows. *Computers & Geosciences*, *97*, 98-109.

Smets, B., Wauthier, C., & d'Oreye, N. (2010). A new map of the lava flow field of Nyamulagira (DR Congo) from satellite imagery. *Journal of African Earth Sciences*, *58*(5), 778-786.

Smets, B., Kervyn, M., d'Oreye, N., & Kervyn, F. (2015). Spatio-temporal dynamics of eruptions in a youthful extensional setting: Insights from Nyamulagira Volcano (DR Congo), in the western branch of the East African Rift. *Earth-Science Reviews*, *150*, 305-328.

Tarquini, S., & Favalli, M. (2010). Changes of the susceptibility to lava flow invasion induced by morphological modifications of an active volcano: the case of Mount Etna, Italy. *Natural hazards*, *54*(2), 537-546.

---

## Author Comment (AC2) · 1 Jun 2021

We acknowledge these meticulous reviews as they improved the manuscript organization and clarify many aspects of the methods as well as the results. Also, they allowed us to deepen the discussion by pointing out the limitations and the strengths of our approach, as well as our crucial interpretation of hazard maps. Reviewer 2 (Hannah Dietterich) raised a large number of forward questions that are totally justified and very interesting. For example, she asked if other numerical models were used, she suggested to incorporate time (probability of inundation at a location over a given time interval), as opposed to just a conditional probability for a future event, or suggested to provide quantification of probabilities in a given area (for example: what is the probability that the road will be cut), etc... Applying all these suggestions, would require significant additional data and great lengthening of the article, as well as data that are outside the scoop of this study. We find more appropriate and valuable for our study to not add more data here. This will ensure our article to be concise (already at length) and accessible for authorities and non-specialist as well. It is very important to recall that there are limited studies (if any) on lava flow hazard map at Piton de la Fournaise, in comparison to other volcanic centers such at Etna, or Hawaii were numerous articles have already been published on the subject. Our aim in this article is to provide a first version of a hazard map of Piton de la Fournaise, so it may serve as a reference study for future specific research topics on Piton de la Fournaise such as testing other methods to compute the hazard map or provide probability of inundation as function of time etc... (that would fulfill the reviewer's suggestions). Also through this article we aim at providing a clear article that explains our rather simple approach for the civil protection and authorities to understand is and to use it as support for potential land use planning and management. As mentioned in the last section of the discussion: "The presented map is thus also intended to aid and guide stakeholders in developing effective mitigation and land use plans that also take into account the main volcanic hazard, with the caveat that our maps are for a "typical" effusive event."

We hope that the article in this new shape and improved content is now suitable for publication given the changes we provided. All major comments were answered and the text improved as required. These changes include new numbering of the sections by gathering methods and data and providing a clearer results section and a longer and deeper discussion. Figure 5 has been improved. Figure 7 has been improved as recommended by both the reviewers and is now described in more details in the results section and well discussed in the discussion section. Anew figure (figure 8) was added to support discussion. All minor corrections were done.

See attached pdf

Please also note the supplement to this comment:
https://nhess.copernicus.org/preprints/nhess-2020-394/nhess-2020-394-AC2-
supplement.pdf

**Supplement:**

**RESPONSE TO REVIEWS**
Reviewer comments are in black and *our answers are in blue*

*Dear Editors and reviewers,*

*We acknowledge these meticulous reviews as they improved the manuscript organization and clarify many aspects of the methods as well as the results. Also, they allowed us to deepen the discussion by pointing out the limitations and the strengths of our approach, as well as our crucial interpretation of hazard maps.*

*Reviewer 2 (Hannah Dietterich) raised a large number of forward questions that are totally justified and very interesting. For example, she asked if other numerical models were used, she suggested to incorporate time (probability of inundation at a location over a given time interval), as opposed to just a conditional probability for a future event, or suggested to provide quantification of probabilities in a given area (for example: what is the probability that the road will be cut), etc... Applying all these suggestions, would require significant additional data and great lengthening of the article, as well as data that are outside the scoop of this study. We find more appropriate and valuable for our study to not add more data here. This will ensure our article to be concise (already at length) and accessible for authorities and non-specialist as well. It is very important to recall that there are limited studies (if any) on lava flow hazard map at Piton de la Fournaise, in comparison to other volcanic centers such at Etna, or Hawaii were numerous articles have already been published on the subject.*

*Our aim in this article is to provide a first version of a hazard map of Piton de la Fournaise, so it may serve as a reference study for future specific research topics on Piton de la Fournaise such as testing other methods to compute the hazard map or provide probability of inundation as function of time etc... (that would fulfill the reviewer's suggestions). Also through this article we aim at providing a clear article that explains our rather simple approach for the civil protection and authorities to understand is and to use it as support for potential land use planning and management. As mentioned in the last section of the discussion: "*The presented map is thus also intended to aid and guide stakeholders in developing effective mitigation and land use plans that also take into account the main volcanic hazard, with the caveat that our maps are for a "typical" effusive event."

*We hope that the article in this new shape and improved content is now suitable for publication given the changes we provided. All major comments were answered and the text improved as required. These changes include new numbering of the sections by gathering methods and data and providing a clearer results section and a longer and deeper discussion. Figure 5 has been improved. Figure 7 has been improved as recommended by both the reviewers and is now described in more details in the results section and well discussed in the discussion section. Anew figure (figure 8) was added to support discussion. All minor corrections were done.*

**REVIEWER 2 : Hannah Dietterich**

**Lava flow hazard map of Piton de la Fournaise volcano**
Chevrel et al.

In this manuscript, Chevrel et al. present an eruptive history database and methodology to produce numerical lava flow simulations and a probabilistic assessment of lava flow inundation hazard at Piton de la Fournaise. Their method builds on previous work done by these authors at other volcanoes to incorporate the complex spatiotemporal eruptive history of Piton de la Fournaise and, as they importantly highlight, the impact of topographic change at a frequently eruptive volcano on evolution in lava flow hazard. Overall, the manuscript is very good and will be of interest to a broad audience. The data, methods, and analysis are all appropriate, but some elements of the methodology and key discussion points are missing or wanting of more details. Below I summarize some major comments, questions, and suggestions. Pardon the length, these comments are largely minor, this is just a topic I am very interested in. There are also a number of typos and word order errors that just require a close reading with fresh eyes by the authors to fix.

**Major comments**

*Using recent eruptions to assess methods and implications*

A unique and powerful element of this work is the assessment of lava flow hazard through time using DEMs and eruptive history data that evolve over recent decades. Although these are used to generally state that changing terrain impacts hazard mapping and that recent flows generally occurred in areas that were previously deemed more likely, a missed opportunity is to use these maps and data to do a full hindcasting assessment and validate the method being presented. How well does the map based on data up to 1997 work for eruptions after 1997? Can this be quantified (e.g., Bevilacqua et al. 2017)? Is it better initially and then gets worse as topography changes more and more? What about up to 2009? This study has the distinction of allowing this critical discussion of how well we expect maps like this to do, what timeframe they're useful for, and where they fail given the availability of data over time and very frequent eruptions. These are topics that are all touched on but they are not framed in this fundamental way, where a hindcasting assessment would organize and strengthen the key conclusions. Showing that the method works well for recent decades would demonstrate validity and support further applications. I recommend adding a section to the discussion on this ("Validating hazard mapping with recent eruptions") and accompanying Fig. 7 with quantification of how well the map did for later flows through time. It is also regularly mentioned, but never demonstrated how topographic changes impact hazard assessment. Since this is a major conclusion of this work, and testable with the time series of hazard maps and eruptions, this is a good opportunity to show the changes caused by topographic evolution with visuals.

*Accordingly, a new section in the discussion was added (section 4.1). We did not complete further analyses to quantify how well do the maps predict future eruptions but we instead discuss this matter in more details as recommended and improved figure 7 accordingly. We also provide one more figure (figure 8) to show the difference of hazard probabilities together with the differences in topography.*

*Treatment of spatiotemporal variability in eruptive behavior*

As mentioned and assessed throughout the paper, there is significant variability in the frequency and style of eruptions both spatially and temporally at Piton de la Fournaise, including rare rift zone events, frequent (but episodic?) summit activity, and cyclic patterns in eruption location and magnitude in recent decades. Currently, these are integrated empirically into a conditional hazard map (probability of lava inundation in the next eruption) that incorporates historical data or geologic mapping at various timescales depending on location. However, in the discussion it is then stated that this approach does not integrate all magnitudes or timescales depending on what the "next" eruption is. Certainly, eruptive history that is left out of the input data for hazard mapping will not be represented (e.g., lava lake activity that is not well preserved in the geologic or historical records), but it is unclear why the authors also do not feel that the map represents events that are included (e.g., recent large eruptions that end each eruptive cycle).

*Indeed this part was re-written in the discussion (section 4.1) as follow:*
*"Because our hazard maps are computed with a database in which only four of the 137 eruptions since 1931 are high volume, source-related-cycle terminating, events (i.e. 1931, 1961, 1986 and 2007), such infrequent events have a low probability and hence may occur in low probability areas. For example, it is clearly visible that the April 2007 lava flow occurred in a low (<0.5 %) probability zone of the 1997 hazard map (Figure 7a). It is therefore important to recall that low probability does not mean that an event cannot happen, it only means that it is less probable, i.e., it is atypical if it happens in a low probability area."*

If the input data are not representative, it would be helpful to offer suggestions for how they could be represented moving forward, or utilize a different methodology for the treatment of these spatiotemporal patterns. Similarly, if the results have limited utility to specific time periods or scenarios, incorporating those explicitly, such as producing different conditional probability maps for specific conditions or regimes (e.g., just the large events that occur only after smaller ones) or a map of probability of any lava inundation over a given time period (1 year, a decade, given that the recurrence rates are well characterized), would help in the discussion. Overall, it seems like the map in Fig. 6 is perhaps more representative of the current state of the volcano, and applicable on the scale of years to decades, than the text implies, because of the relatively complete recent record.

*Yes exactly, this was implemented accordingly.*

In terms of discussing the results, adding/reorganizing the discussion around the "Impacts of cyclicity (or spatiotemporal variability in eruptions more generally) on hazard assessment" instead of methods-based section headings (e.g., 8.1.2), would help focus discussion on a key topic that repeatedly is mentioned throughout the other sections. These cycles are introduced in the background section and talked about a lot in the discussion as changing in location and magnitude through time, but the reader is not shown what these look like or exactly how their properties evolve. This discussion should therefore be accompanied by a figure that shows this well. Maybe marking vent locations and flow extents by cycle position (in different colors) in map view and comparing this to hazard mapping results.

*The effect of activity cycle is already well discussed in the discussion section 4.2 "Historical and geological records: representativeness of future eruptions".*
*We modified the title of a discussion section into : " Accounting for spatiotemporal volcanic activity patterns" as recommended (now section 4.3)*

*Adding a figure about activity cycle is not the aim of the paper and already presented in Derrien 2019. Here we only wish the alert the reader on this fact but not quantify it. For this, a new article would deserve to be written.*

*Methods questions*
Input data for hazard mapping includes vent locations, as well as statistics on the number of lava flows, their lengths, etc. However, these are not clear in the methods given the potential for a given eruption to produce vents that are actually lengthy fissures, multiple vents, and multiple lava flows. Some clarification on how vents, flows, and locations were defined, any implications of these choices on the results, and references to literature on these challenges (e.g., Cappello et al. 2012; Runge et al. 2014), are needed. Given some uncertainty in how these may be defined, the high precision and number of significant figures in the tables seems overestimated and should be discussed.
*The methodology to count vents and lava flows has now been clarified in section 2.3. And the numbers in the table were rounded.*

For the application of DOWNFLOW, were these simulations run from a single vent or from all vents and/or a fissure geometry (lots of point sources along a line, say)?
*From single vents as already mentioned in the methods*
How were ocean- limited flows treated in defining the lava flow length distribution (line 305)?
*We consider here the maximum length, therefore from the vent to where the flow ends (i.e. the coast).*

The DEM resolution is very important to the performance of DOWNFLOW, and separate calibrations are appropriate, so please provide the resolutions when describing the methods (e.g., line 287) and in the discussion of the results (line 453).
*Done*
 In Fig. 4a, it is not clear that the calibration was completed though, given the best-fit seems unconstrained in the parameter-space (Δh >5?).
*There were no need to test Δh >5 because the fit was already good for Δh >4.*

How do multiple vents affect DOWNFLOW modeling/misfit?
*Multiple vents are not considered with this methodology.*
For long flows, the fit seems much worse (Fig. 4a) - greatly overpredicting the inundated area. Would a different model work better for these?
Were other models considered?
*Yes, it is quite intriguing at Piton de La Fournaise that the distal part of the flow is not well fitted by the model. We could have used two different Δh to better fit the distal part of the flow (high Δh for the proximal part and small Δh for the distal part), but here we preferred to keep the simulations straightforward and neglect this by using only one Δh and assuming an overestimation of the distal part.*
*No, other models were not considered. Here we aimed at applying DOWNFLOW.*

 Can uncertainty from this type of misfit be propagated in this methodology?
*Probably but his was not tested here because the scope of the paper was not to test different methodology.*

Additionally, the impact of DEM resolution is worth testing with regard to varying Δh between time periods/DEMs, rather than attributing all differences to thickness changes (e.g., 455). Coarser DEMs (such as the 1997 DEM vs. the >2008 DEMs) have built-in flow spreading from pixel size, potentially greater DEM uncertainty, and significant effective smoothing, which generally seems to change the best-fit Δh significantly. If you resample the later DEMs to 25 m instead of 5 m, does the best-fit Δh change? There may be changes related to flow thickness, too, but DEM resolution should be integrated into this discussion (also in terms of validation – does the 25 m data perform worse?).
*Yes, this was also pointed by reviewer 1 and it is now explicitly discussed.*

*Quantitative analysis of hazard probabilities*
There are a number of places where results are described, but using inundation values for a given pixel, rather than integrating the results to answer the broader questions, such as "what is the probability of lava inundation in the Enclos during the next eruption?" (line 22)
*Direct integration (sum) of the values of the hazard map inside the Enclos will not give the probability of lava inundation in the Enclos because a single lava flow will affect many pixels. However, the answer is actually given in Table 1 where one can read that there is 97.87% of chance that the next eruption will be in the Enclos (23.55 % in the summit crater and 74.16% in the rest of the Enclos).*

or "what is the probability that the next lava flow will intersect the coastal road?" (line 339, this is written as if integrated, but it's just the probability within a given pixel, not along the whole road). The actual probabilities are possible to calculate with these data though (the first is dictated empirically, but the second could be calculated based on the model results).
*We actually cannot get the result by simply integrating along the road because one single eruption does not cut the road in a single point. For this we have to identify all the possible vents that can cut the road and their probabilities to reach the road (e.g. as done in figure 12 of Favalli et al., 2012) and multiply this probability at each possible vent by the probability of having the next vent at that position (e.g. figure 3b of this article) and finally integrate over the region of the possible vents.*
*This would be too much to be added in this paper and we therefore rather not add this information here in order to keep a concise article.*

**Line-by-line comments**
33 – Many lava flow hazard maps also incorporate time (probability of inundation at a location over a given time interval), as opposed to just a conditional probability on the condition of an eruption occurring (e.g., Bebbington 2013; Cappello et al. 2015). Given the known recurrence rates in this study, this extension would potentially be possible here as well, and relevant for applications to land use planning.
*Yes this extension would be potentially possible. However, we do not include this is this article because it would require important lengthening of the manuscript. This will likely be presented in a future article where we will address directly risk maps for mitigation and land use planning.*

64 – "complete" – that seems risky statement given how extensive burial is also described
*Complete has been replaced by "large"*

86 – "Morphological" – replace with "Geological"

*Done*

120 – Introducing these here as low frequency, high impact events will set up why they are so difficult to forecast in the discussion. They're in the data, but they are not the "most likely" event, and less spatially concentrated.
*Yes, you are right. The sentence was reformulated and moved to section 1.3 .*

150 – Expand this to fully introduce the spatiotemporal patterns in eruptive history. Given that episodes on the order of centuries are also later invoked for the methods of database assembly and the short/long-term applicability of the analysis, these could also be introduced here in this background section.
*We believe that the spatiotemporal patterns in eruptive history is already well describe in this section.*

192 – Using the longest flows only will overpredict flow length and yield higher probabilities overall. Perhaps add more explanation (as mentioned in the 'methods' questions above).
*Maybe this part was not clear enough. If the fissure opens perpendicular to the slope, many little flows will propagate until the eruption concentrate on one spot. But if the fissure is parallel to the slope then where ever the lava come out, it will form only one stream (parallel to the slope). To avoid generating errors in number of lava flow, if the fissure is perpendicular and there are many little lobes, we decided to only count one flow per fissure, and to consider the longest one.*
*This is now rewritten as follow: "Note that in the case of a fissure opening perpendicular to the slope, the lava may erupt uniformly along the fissure to feed several lava flow units simultaneously to form a flow field of many lava fingers (Harris and Neri, 2002; Kilburn and Lopes, 1991, 1988). In such a setting we counted only the main, longest flow, and do not consider all fingers that comprise the compound lava flow field in the database (cf. Walker, 1973)."*

Tables 1 and 2 – Are all flows mapped or are some flows that are unmapped (or poorly mapped) but were reported included in these counts? I think since these are directly compared to each other and they have the same rows, it would be much easier for the reader to just combine them into one table.
*Table 1 reports the number of lava flows, while table 2 reports the number of scoria cones. The number of lava flows is restricted for the flows since 1931, while the number of scoria cones is unlimited in time, but consider all scoria cones found on the edifices. To avoid misunderstanding, they cannot be combined in a single table.*
*In contrast in Table 3, we reports counts of lava flows and scoria cones for the same period of time, in that case they are therefore in a single table.*

236 - Is this every single cone, spatter rampart, and fissure? Or grouped/summarized somehow?
*We do not make a difference between single cone, spatter rampart, and fissure. We counted morphologically distinguishable scoria cones as well as vent location for any lava flows.*
*(see section 2.3)*

242 – What function is used for bandwidth, is it symmetric or asymmetric (or fissure geometries) and why?

*We only considered points as possible input vents and used a 'symmetric' Gaussian smoothing function.*

*This because we have many different rift directions, etc., it does not make sense to use an asymmetric function as it would have to vary continuously from point to point. Also we have a great number of input data (compared to other volcanoes) and the rift pattern emerges clearly without 'forcing' them with highly-spatially-variable asymmetric smoothing functions. With the great variability in the density of cones we opted for a bandwidth that is function of the local vent density.*

*We used a Gaussian smoothing function as it is stated in the previous line. We added the word "symmetric" for clarity as follow: "The vent density distribution (number of vents per unit area) was then obtained by applying a symmetric Gaussian smoothing kernel to the map of vent locations (Bowman and Azzalini, 2003; Favalli et al., 2012; Richter et al., 2016), with a bandwidth that is a function of the local vent density (Fig. 3a).."*

Section 7 – The first sentence sets up the goal of 'the next effusive eruption", but this is also making retrospective maps for other timespans, so make sure the plan of making multiple hazard maps that represent different times to look at any changes over time is introduced at the start of section 7. (will make line 323 less confusing on the first read).

*This part was re-written*

Sections 7.1 and 7.2 are interpretive and should be in the discussion. 7.1 can become 8.1 and describe the results, including the integrated example probabilities mentioned above. 7.2 can be modified/replaced into the core validation exercise, comparing the maps and subsequent flows more quantitatively/explicitly to show how well the method works.

*We do not completely agree, Part 7.1 (now section 3) is the core result of this article. We rather keep it into the results and have therefore re-organized the manuscript to make it clear (section 3 is dedicated to results and section 4 to discussion)*

Overlaying flow outlines directly on the hazard maps may be especially helpful. Where it works/doesn't work will then set up the following existing/re-organized discussions of spatiotemporal cyclicity, short/medium/long-term applications, importance of topographic updates, etc.

*Done*

334 – Clearly state that these are conditional probabilities

*Done. This was re-written as follow: " The map clearly shows that, for the given data set, the highest probability of lava flow inundation for the next eruption at Piton de la Fournaise is located within the Enclos."*

Section 8.1.1 - Overall, a 'long-term' hazard map using present-day topography in a frequently active area is most useful on the scale of the next few decades. A short-term forecast map (hours to a month, say) should be using short-term vent opening probability map integrating monitoring data, not the whole volcano over centuries (and up-to-date topography). A century- scale map needs to incorporate the potential of the volcano to change regimes

entirely (it's been stated that eruption frequency can vary spatially and temporally over centuries at Piton de la Fournaise) and the present-day topography is expected to change over time (although it will likely be similar for a long time, large-scale subsidence and such can happen in addition to lava emplacement). So maps like this are most useful in that next years-decades period mostly useful for annualized to lifetime-scale planning and risk, but not shorter or longer term. Products for these other timeframes can also be produced with the same methodology, just with different input data, and that is worth saying here.
*Yes exactly, this was already stated, but it is now clarified.*

425 – This contradicts line 303 (although it's broadly true of course!)
*Yes indeed, this is now better explained.*

429–445 – It seems like these events are represented accurately in the input data (infrequent, less spatially confined), and thus the hazard map. That they occur very sporadically at locations that erupt infrequently does not mean the map is wrong (low-likelihood, large-magnitude). Based on this work, are there recommendations for how these should be treated, or maybe presented differently (e.g., as "worst-case" scenarios or similar)? What work would be needed to include lava lake activity in a future hazard map?
*Recommendation are out of the scoop of this paper but we now clearly stated that "Dedicated studies on the probability of occurrence of such high magnitude and intensity, but atypical, events need to be conducted, and a separate set of hazard maps are required to compute where and when such events are more likely to happen. Likewise, our analysis does not consider the poorly studied, but relatively recent (post-1708), long-lasting activity related to overflow from summit lava lakes, as was common between 1750 and 1800, and again around 1850 (Michon et al., 2013; Peltier et al., 2012). Our maps are, though, applicable to the most common effusive event scenario currently encountered at Piton de la Fournaise. However, they must be used and applied with the above caveats in mind regarding the type of activity and effusive event to which they apply."*

471 – A good place to include some numbers of the integrated probability of trails being inundated in the next eruption, or visually showing the exposure on the map.
*In this work we cannot give integrated probabilities but can only visually show the exposure on the map as we do in figure 6.*

477 - Can you reconcile this? It is an aspect of drainage/integration by accumulation from many possible vents, but it's extremely nonintuitive to readers and the public. Would it be valuable to make a "proximal" hazards layer, say, using a vent opening contour to define a proximal hazard region subject to tephra, ballistics, gasses and near-vent lava flows?
*Actually, there is nothing to reconcile, it is not antagonist, it is just a matter of geometry. Any pixel close to the summit (high altitude) have a contributing area (possible vent location from which the lava path would reach the pixel in question) that is much smaller (up to 1000 times) than for a pixel that is at lower altitude.*

*"a proximal hazard region subject to tephra, ballistics, gasses" is completely out of topic for this article.*

487 - Explicitly highlight that in a short-term or atypical scenario, this framework allows

updating of the input data (DEMs, vent locations, flow properties) to quickly produce a probabilistic map or specific flow forecast scenario as needed.
*Done*

Fig. 6 – I don't see green or white lines. Perhaps adjust line widths and colors?
*This was a mistake and it is now corrected*